



# Capture of near-critical debris flows by flexible barriers : an experimental investigation

Miao Huo[1,*], Stéphane Lambert[2,*], Firmin Fontaine[2], and Guillaume Piton[2]

[1]Sichuan Agricultural University, College of Water Conservancy and Hydropower Engineering, 625014 Ya'an, China
[2]Univ. Grenoble Alpes, INRAE, CNRS, IRD, Grenoble INP, IGE, 38000 Grenoble, France
[*]These authors contributed equally to this work.

**Correspondence:** Huo Miao (huomiao@sicau.edu.cn)

**Abstract.** This study addresses the key issue of the interaction between debris flows and flexible barriers based on small-scale experiments for which both the flowing mixture and the barrier were designed to achieve similitude with real situations in Alpine environments. The considered debris consisted of a large solid fraction mixture with large and angular particles, flowing down a moderately inclined flume and resulting in near critical flows, with a Froude number in the 0.9–2 range. The flexible barrier model consisted in 3D printed cables and net. The flow characteristics, evolution and deposition after contact with the barrier as well as the deformation and the loading experienced by the barrier were addressed varying the flume inclination and released mass. Four different interaction modes between the flow and the barrier are identified increasing the flow kinematics. A model based on the hydrostatic pressure assumption reveals relevant for estimating the total force exerted on the barrier when all the released material is trapped. This force doubles in case there was barrier overflow.

## 1 Introduction

Debris flows threaten people and assets in mountainous regions and capturing them with barriers is one of the most effective protection strategy (Piton et al., 2024). Over the last decade, a growing number of articles have focused on the interaction between debris flows and both rigid and flexible barriers (see e.g. the reviews by Poudyal et al., 2019; Vagnon, 2020b). The change in flow dynamics in the barrier vicinity and the loading exerted by the flow on the barrier have been widely addressed based on numerical simulations, field experiments and on small-scale experiments.

A key dimensionless number driving the various regimes of impact with obstacles such as barriers is the Froude number $Fr$ computed as (Faug, 2015, 2021; Laigle and Labbé, 2017; Vagnon, 2020b):

$$Fr = \frac{U}{\sqrt{\cos\theta g h}} \qquad (1)$$

with $U$ the velocity of the flow front [m/s], $\theta$ the channel inclination [°] (note that for mild slope, e.g. for $\theta<15°$, $\cos\theta > 0.96 \approx$ 1 and is usually ignored in the equation), $g$ the gravitational acceleration [m/s²], $h$ the front thickness (hereafter referred to as the 'depth') measured perpendicular to the flume bottom.

The Froude number of debris flows observed in the field is variable depending on the flowing material and channel characteristics. Many references dealing with debris flows in Alpine environments suggest Froude numbers ranging from 0.5 to



2 (Costa, 1984; Hungr et al., 1984; Jacquemart et al., 2017; Wendeler et al., 2019; Nagl et al., 2024). This typical range was

recently confirmed by McArdell et al. (2023) and by Lapillonne et al. (2023), based on direct and accurate monitoring of 35 debris flows at the Illgraben torrent (Switzerland) and of 32 debris flows at the Réal torrent (France), respectively.

Debris flows with Froude numbers >2 – 4 exist in nature but well documented cases appear to correspond to particular contexts. Based on a direct monitoring on the Mt Sakurajima volcano, in Japan, Watanabe and Ikeya (1981) reported $Fr$ ranging within 1.0 – 2.7. Theses values relate to flows referred to as "lahars" in which volcanic ashes induce a lubrication effect

resulting in flows faster than usual debris flows. Mostly supercritical surges were also measured in the peculiar catchment of the Jiangjia Gully in China (Hu et al., 2013; Guo et al., 2020, 2024), typically in the 1.5 – 3.5 range, with some surges >4. Nevertheless, comparison with other catchments for which data from mud-flows and debris flows monitoring are available reveal that the Jiangjia Gully experiences rather fast debris flows (Phillips and Davies, 1991; Lapillonne et al., 2023; Guo et al., 2024). Consistently, based on a specific flow velocity estimation approach, Prochaska et al. (2008) reevaluated previous field

observations concerning debris flows and reached the conclusion that $Fr$ rarely exceeds 3.5, an upper bound already visible in the data compiled by Phillips and Davies (1991). Froude number exceeding 4 are sometimes mentioned in the literature. One of the most frequently cited very high $Fr$ value originates from Fink et al. (1981) who back-computed the features of two surges in the very steep Pine Creek (gradient up to 30°) on the Mount St. Helens. The velocity of the two surges was reconstructed from deposits in bends and resulted in a Froude number >8 for a super fast (15–31 m/s) and very big surge (with estimated

peak discharge = 2800 – 3400 $m^3$/s) in a very steep reach (gradient = 17–30°). This surge had the features of lahars and its Froude number decreased down to $\approx 2$ further downstream where the gradient was 4°. Although fast and shallow debris-flow surges resulting in high Froude numbers can be reported, also in case of very diluted mud-flows (Yune et al., 2013; Kim et al., 2023), they should be considered peculiar and rather exceptional as related to specific conditions in terms of flowing material characteristics and steepness in particular.

Among the numerous works dedicated to the investigation of the impact of debris flow on structures, the vast majority nonetheless considered highly supercritical flows (i.e. $Fr$>1), with $Fr$ ranging from $\approx$2.5 to >10 (Bugnion et al., 2012; Canelli et al., 2012; Ashwood and Hungr, 2016; Cui et al., 2018; Shen et al., 2018; Ng et al., 2019; Tan et al., 2019; Vagnon, 2020a; Li et al., 2020; Song et al., 2021; Kong et al., 2022; Kim et al., 2023; Xiao et al., 2023; Berger et al., 2024). On the contrary, debris flows with a $Fr$ <2.5 have been much less considered when dealing with mitigation structures (but see Scheidl et al.,

2013, 2023; Ashwood and Hungr, 2016; Chehade et al., 2021; Song et al., 2023). In the end, it appears that most of the published findings which serve as basis for improving design methods of mitigation structures concern supercritical to highly supercritical flows. As yet suggested by Hübl et al. (2009), it can be considered that models in relation to the flow-barrier interaction have globally been developed on an input data range which does not comply with the most frequently observed debris flows type in Alpine areas. This difference in Froude number results in a difference in loading regime on the barrier.

Indeed, many authors have evidenced that above a Froude number of $\approx$1.4, the loading exerted by a granular flow onto an obstacle was dominated by inertia forces, which relate to the flow velocity, and that, below this value, the loading on the barrier was mainly dominated by gravity forces, associated with the depth of the intercepted material (Tiberghien et al., 2007; Laigle and Labbé, 2017; Wendeler et al., 2019; Huang and Zhang, 2022). In other words, the difference in $Fr$ between most research





conditions and what is observed in nature leads to an excess in the attention on the influence of the flow velocity, which has
consequences on the knowledge as for the way the flow accumulates, deposits and overflows the structure and more generally
interacts with it. There is thus a vital need for an in-depth investigation of the flow-barrier interaction while considering debris
flows with a Froude number closer to that in most frequent real cases, i.e. <2, in particular in view of improving the design of
mitigation structures such as flexible barriers.

The design of flexible barriers intended to intercept debris flows is classically conducted by modelling the loading it expe-
riences as a combination of a static component on its lower part and a dynamic one above (Wendeler, 2008a; Ng et al., 2012;
Ferrero et al., 2015; Sun and Law, 2015; Song, 2016; Tan et al., 2019; Berger et al., 2021). The former corresponds to the
loading exerted by the material deposited upstream the barrier (also referred to as 'backfill' or 'dead zone') and it is modelled
as an hydrostatic pressure along the deposit depth. The latter component corresponds to the action due to the flowing material
and is computed as an hydrodynamic loading accounting for the debris flows velocity and density and for an empirical dynamic
pressure coefficient of 2 for granular flows and in the 0.7-1 range for viscous flows (according to Berger et al., 2021). Different
scenarios are accounted for in design recommendations, in particular in terms of dead zone height (i.e. non-flowing area), bar-
rier overflow, number of surges with consequences on the respective contribution of these two components on the barrier total
loading. As most of the research conducted up to now concerned high $Fr$ values, findings from the literature mainly concern
high velocity flows, and limited attention was paid to the static component associated with the deposited material.

All in all, this study concerns near critical debris flows (i.e. with $Fr = 0.9 - 2$), and involves flume experiments where
flexible barriers in mechanical similitude with the real-scale are used as mitigation structures. The study is conducted varying
the flume inclination and the mass of released material while considering different barriers. A particular focus is placed on
the flow description and on the barrier loading at rest, which can be assimilated to the static loading component considered
in design practices. The paper is organised as follow: (i) the Material and Method Section describes the flume setting, how
the debris flow material was prepared and experiments were run, as well as the experimental plan. (ii) The Results Section
describes the features of the approaching flows, the interaction modes and the barrier loading. A discussion and a conclusion
close the paper.

## 2  Materials and methods

The laboratory experiments were carried out in an inclined flume to extremity of which the flexible barrier was secured. A
coarse and saturated mixture of grains, clay and water was released from a reservoir into the flume and subsequently reached
the barrier. The characteristics of the flowing material and of the flexible barrier were designed to meet similitude criteria, at a
1/40 scale. Various equipments were used to characterize the flow propagation and the flexible barrier response with time.

### 2.1  Flume

The flume was 5.6 m in length, 0.3 m in width and 0.5 m in height and had one lateral transparent wall (Fig. 1). The flume
bottom and lateral wall were flat and smooth. Its inclination could be varied from 10° up to 20°.



To prevent from consolidation, a mixing facility into a cylindrical reservoir was installed at the upstream extremity of the flume (Fig. 1a). The debris flow material was mixed by mixers in this reservoir and then released in the flume by manually opening the butterfly valve underlying the reservoir.

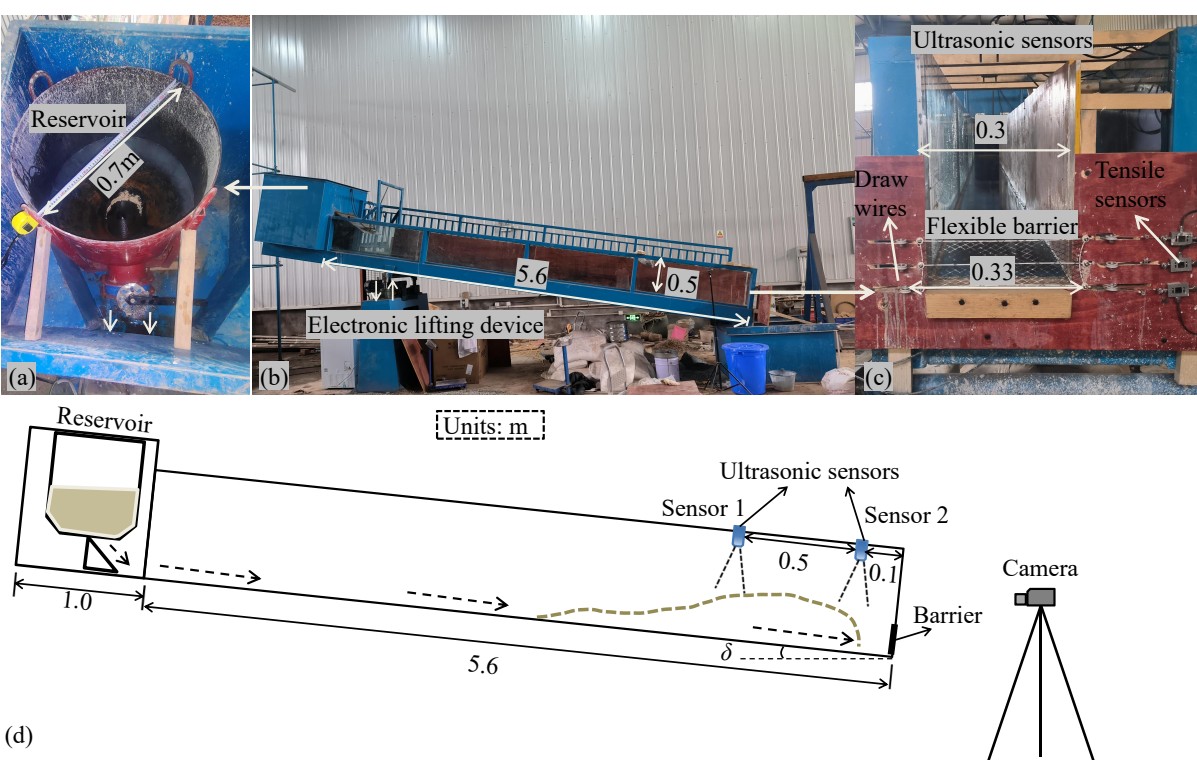

**Figure 1.** Experimental facility: (a) reservoir for initiation, (b) side view of the whole flume, (c) frontal view of the flexible barrier model and (d) sketch of the flume model.

## 2.2 Flowing material

The flowing material consisted of a mixture of sand gravel, clay and water with a solid fraction consistent with real debris flows (76% of grain, 24% of water and clay). The characteristics of the mixture (grain size distribution, water content) were determined to mimic coarse-grained debris flow (high solid fraction) while considering a 1/40 scale ratio (driving for instance the maximum grain size that would be 0.8 m at prototype scale) and meeting similitude requirements thanks to the Froude number, (i.e. $Fr \approx 0.9 - 2$).

Preliminary tests were conducted to determine the mixture content in angular coarse (10-20mm), medium (1-10mm) and fine (0.05-1 mm) sands, as well as in kaolin clay (< 0.05 mm). These tests were conducted in the flume without any flexible barrier, considering 10 different mixtures and varying the flume inclination while measuring the flow depth and velocity. The optimum mixture was determined so that $Fr$ typically ranged between 0.5 and 2. These preliminary tests resulted in the recipe





presented in Table 1 where grain size distribution reveals a $d_{85}$ value of 15.6 mm (Fig. 2). Noting that 60 % of the solid mass
is larger than 1.5 mm (i.e. 60 mm at the real scale), this material constitutes a high solid fraction with high fraction of large
particles debris flow model (Scheidl et al., 2023).

The minimum Froude number of this mixture flowing down the flume inclined by 11° was measured to be 0.9 approx.
Increasing the flume inclination and total mass of the flowing material subsequently allowed achieving higher $Fr$ values, up to
2.

The solid particles were firstly sieved and then stored in different buckets with specific diameter tags. Both water and particles
with given amounts were put into a big bucket and stirred constantly for an initial mixture, and the mixture was subsequently
divided into several buckets to be lifted manually into the reservoir at the top of the flume waiting for the initiation. A new
stirred mixture was prepared for each test.

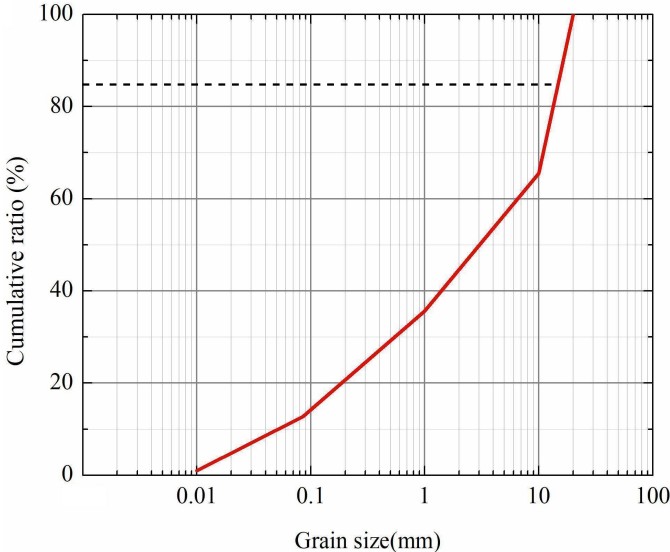

**Figure 2.** Grain size distribution of the flowing material considered in this study (laboratory scale).

**Table 1.** Flow material mixture recipe

|  | Water (%) | Coarse sand | Median sand | Fine sand | Kaolin |
|---|---|---|---|---|---|
| Grain size (mm) | - | 10-20 | 1-10 | 0.1-1 | <0.048 |
| Mass content (%) | 13 | 30 | 26 | 20 | 11 |

This material was mixed until it was released to remain unconsolidated. The flume was cleaned after and before each test by
water flushing to eliminate any solid material (sediment residual).





## 2.3   Barrier

Similarly as for the debris flow material, the barrier model used in the experiments was designed to resemble real structures while considering similitude requirements, in line with a recent research which focused on flow-driven logs trapping (Piton et al., 2023; Lambert et al., 2023, 2024). The model corresponds to a real barrier 13m in lenght and 4 m in height, comprising 120 a net with a water-drop mesh net supported and laterally fringed with 24 mm in diameter steel cables, 150 GPa in tensile modulus. By contrast with real barriers, the model doesn't integrate any energy dissipators, which have a significant influence on the horizontal cables loading and on the barrier deflection (Albaba et al., 2017). To the best of our knowledge, the present study is the very first that addresses the trapping of debris flows considering a down-scaled flexible barrier.

The net model was 100m × 330 mm in dimensions, with a diamond-shape unit mesh, 9.4 mm × 20.6 mm in dimensions (Fig. 125 3). The net was 3D printed from PETG (polyethylene terephthalate glycolized)(see Lambert et al. (2023) for its description). This net was supported by 3 horizontal cables, 330 mm in length and connected to 2 vertical lateral cables 100 mm in length (wing cables). The cables were 3D printed from a Stereolithography resin (type JS-UV-2018-01). Single and pairs of cables were used as horizontal cables, to accommodate to the loading experienced by the barrier throughout the test campaign. Cables made from metallic wires 0.3 and 0.5 mm in diameter were also used as horizontal cables for illustration purpose. The cables 130 were woven in the net to insure mechanical connection. The extremities of the vertical cables were connected to the extremities of the upper and lower horizontal cables. The flexible barrier model was installed at the flume outlet, normal to the flowing direction.

The main characteristics of the barrier models are presented in Table 2. The last column gives the ratio between the model parameter value and the value of this parameter meeting similitude requirements. Due to some manufacturing constraints, 135 perfect similitude is not achieved for some cable characteristics (i.e. cases where the ratio ≠ 1). For example, the 3D printed cable stiffness is too large. In spite of this, it is believed that the barrier model response is representative of that of the real scale prototype. As for metallic cables, similitude was not a requirement but it is noteworthy that the stiffness of the 0.3 mm in diameter steel cable is close to that of the 3D printed cable.

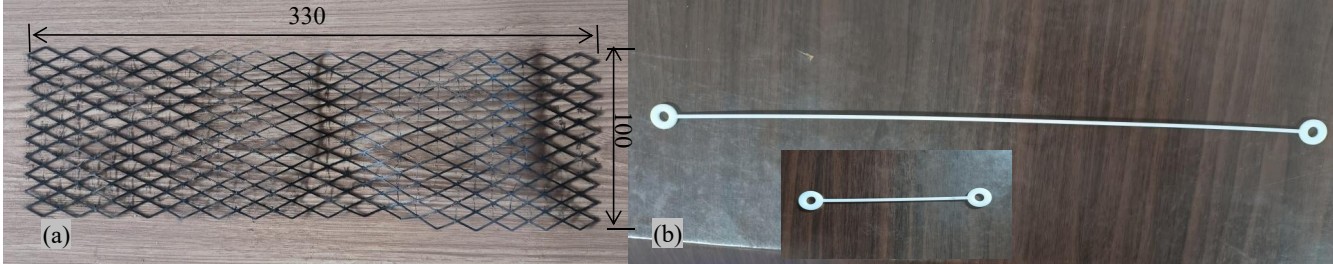

**Figure 3.** Flexible barrier model: net (a) and cables (b).



**Table 2.** Characteristics of the flexible barrier models.

|  | Value | Compliance with similitude |
| --- | --- | --- |
| Barrier length (m) | 0.33 | 1 |
| Barrier height (m) | 0.1 | 1 |
| Unit mesh size (mm) | $9.4 \times 20.6$ | 1 |
| Net stiffness ($*10^4$N/m/m) | 0.6–3 | 1.2–1 |
| Normal-to-the-flow cable transverse dim. (mm) | 1.9 (3D printed cable)<br>0.3 and 0.5 (metallic cables) | 0.5–3.2 |
| Cable stiffness ($*10^3$N/m/m) | 5.8 (3D printed cable)<br>6 (metallic cable 0.3 mm)<br>14 (metallic cable 0.5 mm) | 2.7 (3D printed cable)<br>2.8 (metallic cable 0.3)<br>6.4 (metallic cable 0.5 mm) |

## 2.4 Measurements

Equipment used during the experiments aimed at measuring the flow velocity and depth upstream the barrier, the barrier elongation along its length and the force in the barrier horizontal cables.

Ultrasonic sensors (US) were installed 0.5 m above the flume bottom with their main axis normal to the flume bottom. Sensors 1 and 2 were respectively installed 0.6 m and 0.1 m upstream from the flume extremity. The collected data were used for determining the front velocity and the depth (or height) of material above the flume base during the test (see Section 2.5).

Two cameras were installed on the side and in front of the channel to record the profile and frontal views of the flow-structure interaction. The side views show the scene from a 11 cm distance from the barrier approx. These images were used to analyse the flow evolution during its interaction with the barrier (see section 3.2.1) and also for providing an estimate of the maximum depth.

The three horizontal cables were equipped with force and elongation sensors. More precisely, the eyelet at one extremity of 150 each cable was connected to a force sensor with a 1000N capacity. Elongation of the barrier was recorded using displacement sensors which wire ran along the barrier cable to which it was secured, in a similar manner as in Piton et al. (2023); Lambert et al. (2023). The precision of the displacement sensors was as small as $1 \times 10^{-4}$ mm.

Dataloggers with a sampling rate of 100 Hz recorded the ultrasonic and draw wire sensors measurements. Synchronization between time series were performed based on time of impact defined based the elongation measurements.

## 155 2.5 Data post processing

Data collected during the tests were post-processed to derive the flow velocity and loading on the barrier.

The approaching flow depth depth, $h_f$ was determined from data collected from US1. Data from US2 aimed at sensing the surge incoming right before impacting the barrier.





The approaching debris flow front velocity, $U$, was determined from the time lag between the two ultrasonic sensors which
were 0.5 m apart (Vagnon and Segalini, 2016; Hürlimann et al., 2019; Ng et al., 2019). Given that this type of sensor measures
the distance within a cone, contrary to laser sensors for example (Wang et al., 2022), the collected data were post-treated with
the Pearson correlation method to obtain the time lag. The principle is basically illustrated in Fig. 4. To cross control that the
time lag was correctly assessed, the downstream signal is also shifted in the figure by the computed time lag. If the continuous
increase part of US1 correctly align with the shifted part of US2, it means that the time lag is coherent. In the rare cases where
the automated computation of the front velocity was not consistent (misaligned dotted pink line with the continuous blue line),
a verification and estimation based on the signal and videos was performed.

The force sensors were mainly used to determine the force at rest, after the material has stopped flowing. Indeed, obtaining
a precise measurement of the force during the event was not possible due to some technical limitations with the system. The
measures collected from these sensors will be used in section 3.3 for estimating the total force transferred to the barrier anchors
(force within the barrier) and for giving an estimate of the total force exerted by the debris flow, as detailed in Appendix A.

Video images from the front were used to assess cases that resulted in overflow and also allowed identifying barrier failure
cases. Video images from the side were used for assessing the flow profile evolution during the test. These videos images were
also used to derive the maximum depth of the material during the event, $h_m$, and the maximum depth of the depth material
once at rest, hereafter referred to as deposit depth, $h_d$. Indeed, comparison with the ultrasonic sensor measurements sometime
revealed unreliable due to the deposit surface shape, which was often not parallel to the flume bottom. In addition, this surface
was uneven, with a significant roughness due to the largest particles size. This introduced uncertainty in the depth measured
from the US, as well as from the videos.

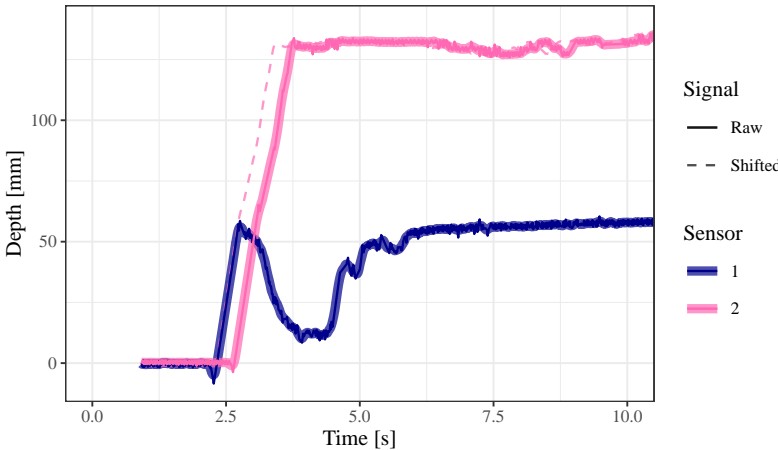

**Figure 4.** Flow depth time series (raw signal in continuous line) and shifted downstream measurement (dotted line) to cross check that the
time lag deduced from the correlation analysis correctly align the flow level of the surge front





## 2.6 Experimental plan

This study focuses on the interaction between a single surge debris flow and a flexible barrier. In view of accounting for different
flow dynamics, the total mass of the released mass of mixture, $m$, and flume inclination, $\theta$, were varied. Some combinations
of $m$ and $\theta$ resulted in cable failure while the material was still flowing. This in particular occurred with $\theta \geqslant 14°$ and/or
$m = 200$kg. The corresponding data were discarded from the study. In order to conduct tests at large $m$ and $\theta$ values, the
flexible barrier design was modified adding one supporting horizontal cable at each position. The tests considered in this study
are listed in (Table 3). Each test is named according to a code, e.g. 11-100-3D1 listing the inclination (11, 13, 14 or 15 in
degrees), the mass released (100, 150 or 200 in kg), the type of cable ("3D" for 3D printed or "met" for metallic) and whether
or not the cables were doubled ("3D1" are single 3D printed cables while "3D2" are doubled 3D printed cables).

All tests where 3D printed cables were employed were repeated at least three times. Less tests were conducted with metallic
cables, as these structures were considered in a forward-looking initiative, for comparison purpose and also to address higher
mass and inclination values. A higher number of repetitions was considered for test series 13-100-3D1 and 14-100-3D1. In
the first case, this was due to minor differences in barrier design concerning the lateral cables. In the second case, three tests
resulted in a late and partial barrier failure and, even though this induced very marginal effect on the deposit volume and
shape, additional tests were conducted. These differences were accounted for when analysing the results in view of considering
consistent data sets with respect to the addressed topic (e.g. tests where late cable rupture was observed were considered when
dealing with the incoming flow characteristics).

**Table 3.** Experimental plan

| Test series | Flume incl. | Released mass | Horizontal cable | Number of |
| Id | $\theta$ (°) | $m$ (kg) | type | repetitions |
|---|---|---|---|---|
| 11–100–3D1 | 11 | 100 | single 3D printed | 3 |
| 11–100–3D2 | 11 | 100 | double 3D printed | 3 |
| 11–150–3D1 | 11 | 150 | single 3D printed | 3 |
| 11–200–3D2 | 11 | 200 | double 3D printed | 3 |
| 13–100–3D1 | 13 | 100 | single 3D printed | 8 |
| 13–100–met3 | 13 | 100 | metallic 0.3 mm | 2 |
| 13–100–met5 | 13 | 100 | metallic 0.5 mm | 1 |
| 13–150–3D1 | 13 | 150 | single 3D printed | 3 |
| 14–100–3D1 | 14 | 100 | single 3D printed | 7 |
| 14–100–3D2 | 14 | 100 | double 3D printed | 3 |
| 14–200–3D2 | 14 | 200 | double 3D printed | 3 |
| 15–100–3D1 | 15 | 100 | single 3D printed | 3 |
| 15–100–met5 | 15 | 100 | metallic 0.5 mm | 1 |
| 15–150–met5 | 15 | 150 | metallic 0.5 mm | 1 |





# 3  Results

## 3.1  Approaching flow

Figure 5 illustrates the behaviour of the flowing material before reaching the barrier from front and side views. Very similar trends were observed in other test conditions. First, the front consists of a rather homogeneous mixture of large and fine particles (Fig. 5a). Second, the flow front is clear and rather steep rising from a dry bed to its maximum in about 25–30 cm (Fig. 5b). In all cases, the flow depth remained rather constant over a rather long distance after this maximum value was reached.

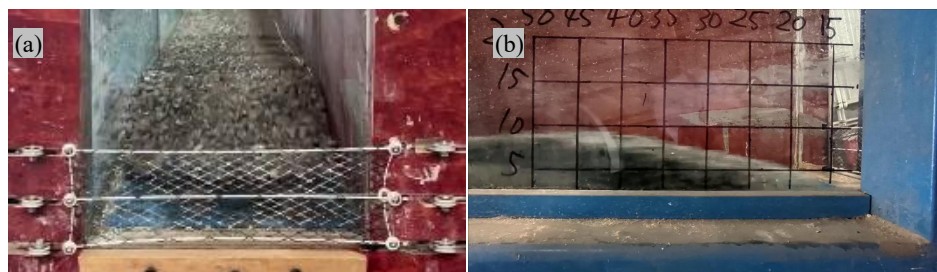

**Figure 5.** Picture from (a) the front and of (b) the side of the flowing material before it reached the barrier (Test 13-100-3D1).

The mean values out of all experiments for the flow front velocity and depth $h_f$ depth were approximately 1 m/s and 7.5 mm, respectively. These characteristics revealed significantly variable from one test condition to another, depending on the released mass and flume inclination (Fig. 6). In particular, the larger the released mass, the higher the flow depth (Fig. 6a). By contrast, no clear influence of the flume inclination on the flow depth can be observed on the considered range. Meanwhile, as expected, a higher inclination results in a higher velocity, for all released mass (Figure 6b). However, there is no clear trend for the influence of the mass on the velocity. In brief, the released mass had an influence on the flow depth while the inclination had an influence on the flow velocity.

In the absence of measurements of dynamic viscosity within debris flow, Reynolds number is not warranted as the scaling index in this study. Froude number, $Fr$, is the dominant parameter to verify the flow conditions and scaling basics (Fig. 7). $Fr$ was computed from the flow velocity and depth measurements and revealed to range from 0.9 to 2.0. The flow regime was thus subcritical to supercritical (Faug, 2021). $Fr$ globally showed an increase trend with the increase in inclination. By contrast, no influence of the released mass on $Fr$ was evidenced.

The box plots in Figure 6 and Figure 7 reveal a variability in flow characteristics among tests conducted with a same mass and same inclination. For instance, releasing a 100-kg mass in the flume inclined by 13°, $Fr$ ranged from 1 to 2, approx. These observations reveal that having a well defined recipe for such a mixture is not sufficient for obtaining the same flow characteristics for a given flume inclination $\theta$ and released mass $m$. We believe that the observed variability is partly attributed to the difficulty, for such a mass of that high solid fraction mixture made of coarse and angular grains, in ensuring the very same state at flow initiation and also to some associated segregation effects. To our experience, this effect is more limited in a steeper flume but it would result in excessively fast flows as yet discussed. This variability in $Fr$ which is also observed in the





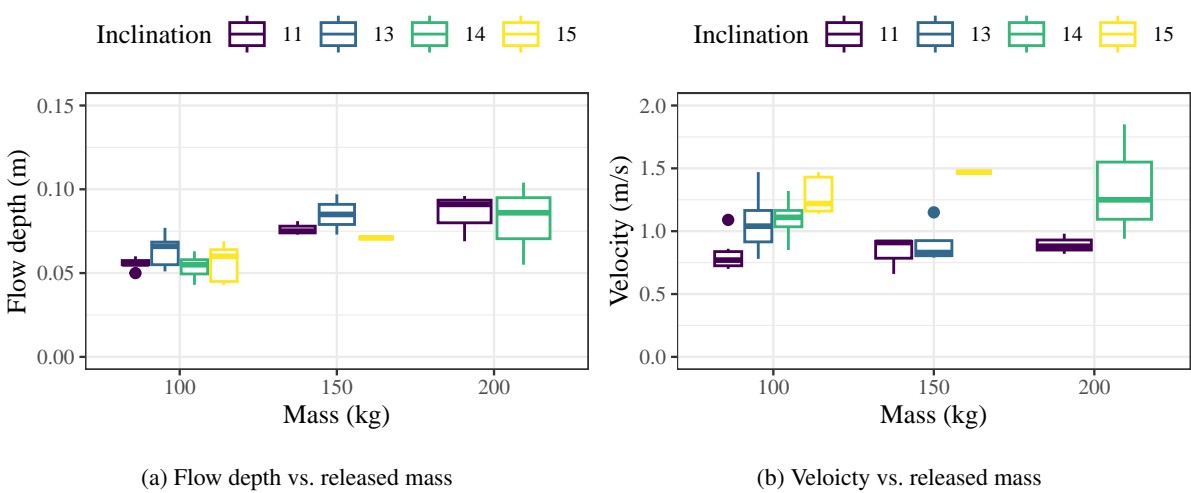

**Figure 6.** Depth and velocity of the flow before reaching the barrier. All test conditions.

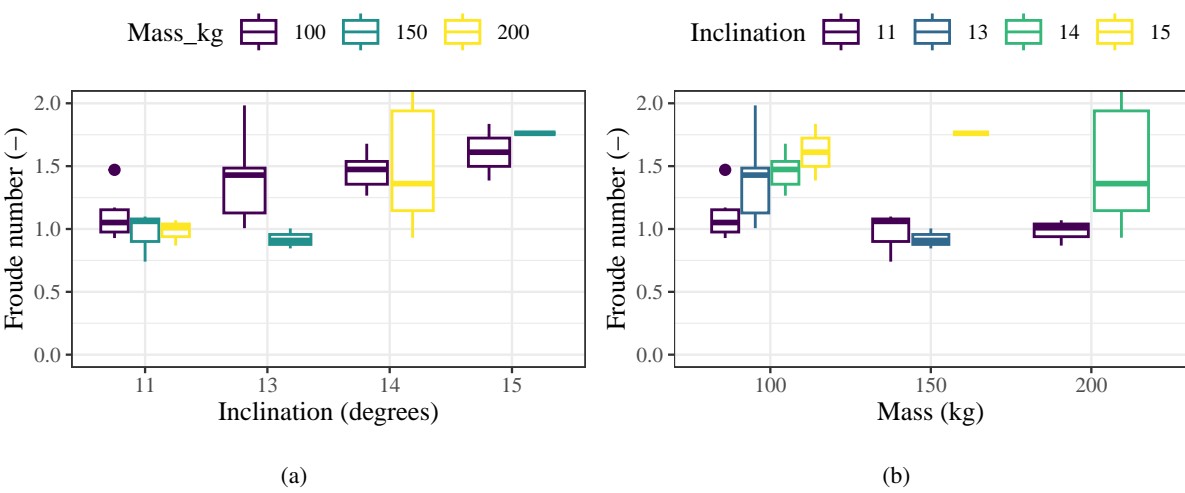

**Figure 7.** $Fr$ vs inclination (a) and released mass (b). All tests conditions.





field within a given stream, clearly justifies conducting test repetitions. In addition, this variability suggests that comparison between test results should rather consider the effective incoming flow characteristics (i.e. approach flow depth velocity and $Fr$) rather than the test conditions (flume inclination or mass).

## 3.2 Flow evolution after contact with the barrier

### 3.2.1 Interaction modes

The influence of the barrier on the flow evolution is generally described according to two interaction modes, referred to as "pile-up" and "run-up", which occurrence depends on the flow characteristics (frictional or viscous) (Armanini and Scotton, 1993; Sun and Law, 2015; Faug, 2021; Ashwood and Hungr, 2016; Kong et al., 2021), which in turn could affect the derivation of the impact force. In essence, the run-up interaction, refers to the flow forming an upward jet along the barrier. By contrast, the pile-up interaction, which is also referred to as momentum jump mode (Albaba et al., 2018; Song et al., 2023), is associated
with the formation of a reflected wave (or granular jump), which is attributed by some academics to a progressive accumulation of material over the dead zone formed close to the barrier. This contrasts with observations made in this study increasing the mass and flume inclination which revealed a gradual change in interaction mode (in terms of accumulation, deposition and overflow) from a gentle one where all the material was arrested quietly and almost instantly to a strong one with overflowing.

The effect of the barrier on the flow was analysed based on the side and front videos. The observed trends are detailed
in the following. These global trends appeared rather independent on the cable types and number. Consequently, the videos analysis focused on the factors in relation with the mass released and flow characteristics which best explained these trends notably through a mass integration of the energy specific head while assuming a uniform flow depth, $h$, – a reasonable first order approximation – which leads to:

$$E = mgh(1 + Fr^2/2) \tag{2}$$

Globally, four different modes could be identified, all described in the following paragraphs. The first was the quasi-solid body behaviour (mode $I$). The second was the granular buckling dominated mode, without overflowing and with limited evidence of granular jump (mode $II$). Mode $III$ was related to cases with a pronounced granular jump but without overflow. Mode $IV$ concerned all cases where the material accumulation upstream the barrier was followed by overflowing.

Mode $I$ consisted in a gentle interaction where the flowing material almost behaved as a solid body at its interception. Once
a volume of material was arrested, it was exposed to compaction by the flowing material, resulting in the arrested material expansion in the vertical direction (Fig. 8a). This process started in the barrier vicinity and concerned almost instantaneously all the released volume, without any flow by the subsequent incoming material over previously arrested material. The vertical expansion was uniform along the flume length. It resulted in a deposit surface almost parallel to the flume bottom, with a depth slightly higher than the approaching flow depth (on the average, 10.5 vs 9 cm resp.). This interaction mode was observed for a
limited number of cases where the flow energy was about 100 J and $Fr$<1.

The increase in flow kinematics (velocity and flow energy) resulted in mode $II$ characterised by a higher expansion of the arrested material combined with a free surface shape change with time. In fact, the depth increase started at the barrier and





propagated upstream as a low to very low amplitude reflected wave (i.e. resembling a granular jump)(Figure 8b). By contrast with mode $I$, it seems that the compression generated by the incoming flow resulted in instability in the accumulated material (mechanism similar to chain forces buckling in granular materials) which explains the higher deformation of the accumulated material. No material flowing over the yet-arrested material at the barrier bottom could be visually observed. As a result of the significant vertical expansion, the maximum depth of the accumulated material, $h_d$, exceeded the barrier initial height, $h_0$, by up to 50 %. During some tests, some particles from the surface of the accumulated material, in the barrier vicinity, were destabilised and felt beyond the barrier. The deposit had a variable depth along the flume length and exhibited a convex shape with a depth in a 10–15 cm typical range. This mode was the most frequently observed and concerned all flume inclinations, velocities and Froude number values, and flow energies over a wide range up to 180 J.

Although their flow energy is similar (i.e. <170 J), more pronounced granular jumps were observed in mode $III$ having flows with a $Fr$ >1.2, which appear mostly at flume inclinations of at least 14°, without any clear relation with other factors (Figure 8c). This mode is assimilated to the momentum jump mode (Albaba et al., 2018) but the way the granular jump formed (e.g. by incoming material piling up above the arrested material) could not be determined visually. In these cases, the depth of material in the barrier vicinity was much more than twice the incoming flow depth.

Massive overflow was observed in mode $IV$ with $E$ in a 200–350 J typical range (Figure 8d). In fact, the occurrence of overflow was more clearly related to the released mass, as all cases with a mass of 200 kg resulted in barrier overflow. One case where the mass equalled 150 kg resulted in overflow. Overflow occurred after a significant volume of material accumulated upstream the barrier, with a well-marked granular jump in some cases (meaning step-like). The depth of the accumulation reached a maximum value, typically twice the barrier height, before significantly decreasing with time as the arrested material was flushed downstream by the subsequent incoming flow. After the test, the deposit depth, $h_d$, was slightly higher than the barrier initial height and the surface of the deposit was nearly parallel to the flume bottom.

In the specific case illustrated in Figure 8e, massive overflow was observed after the flow ran-up along the barrier. This was the only case where run-up with an upward jet was observed. Compared with cases in similar tests conditions, this flow was characterised by high velocity, low depth, high $Fr$ (1.8) and energy above 400 J. This was related to an apparent lower solid fraction of the flow front which resulted from some segregation effects at flow initiation, as mentioned before. For these reasons, this case was considered marginal and thus not representative of the system response in the test conditions ranges.

The analysis of the side videos from the moment the flow front touched the barrier to the situation at rest thus reveals that the interaction between the flow and the barrier was variable depending on the test conditions, in terms of the flow evolution, material accumulation and deposition and barrier overflowing. It also revealed it could hardly be described in a binary way as often done in the literature, where pile-up and run-up are proposed as the two processes by which an obstacle modifies the flow kinematics. On the contrary, a progressive shift between four typical modes was observed. These differences with descriptions from the literature are thought to be related to the differences in flowing material characteristics and in flow conditions, and in particular the focus on near-critical debris flows with a rather narrow range of Froude number.





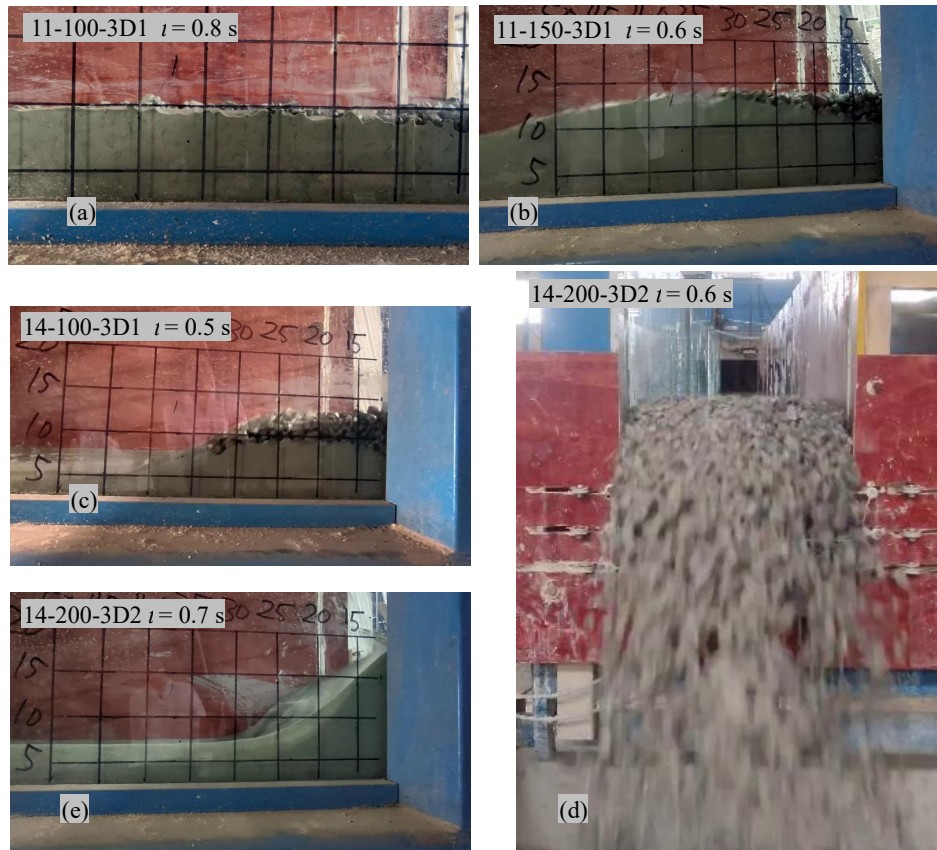

**Figure 8.** Illustration of the different modes observed: (a) quasi-solid body behaviour, mode $I$ (b) granular buckling dominated, mode $II$, (c) pronounced granular jump, mode $III$ and (d) massive overflow, mode $IV$. Marginal case of quasi-vertical jet (e). The time reference (t=0) corresponds to the moment when the flow front touched the barrier.

### 3.2.2 Depth variation and energy dissipation

The increase in depth resulting from the flow interaction with the barrier was addressed based on the ratio $\beta = h_m/h_f$ between the maximal value of the flow depth after it reached the barrier and the incoming flow depth. This ratio was determined for comparison purpose between the different tests, based on the side video images. This ratio was considered by some authors

290 and two analytical expressions relating $\beta$ with $Fr$ have been proposed. The first expression was proposed assuming energy conservation in the case of slow flows (Faug, 2021):

$$\beta_e = 1 + \frac{Fr^2}{2} \tag{3}$$





The second expression was established based on the momentum jump theory which considers an abrupt change in the flow depth during the impact (Armanini, 2009):

$$\beta_m = (1 + 1.5 Fr^{1.2})^{\frac{5}{6}} \tag{4}$$

Both equations rely on the hypothesis that all the material is stopped by the obstacle which is consistent with our experiments except for the mode $IV$ results.

As shown in Fig. 9, experimental $\beta$ values ranged from 1.2 to more than 3.3. The observed scattering is mainly attributed to the variability in flow characteristics. In spite of this scattering, some general trends can be observed. The positive correlation of $\beta$ with $Fr$ is basically explained by the fact that increasing the flow velocity induces an increase in accumulation depth during the interaction of the flow with the barrier. This figure also reveals a dependence on the interaction modes shown in Fig. 8. Interaction modes $I$ and $II$ globally result in lower $\beta$ values, even at relatively high Froude numbers. Measures related to mode $II$ cases are significantly scattered. Higher values of $\beta$ over the whole range of Froude number are associated with mode $III$ and, to a slightly lesser extent, with mode $IV$. This suggests that higher depths were reached during mode $III$ cases, because there was no overflow.

This figure also shows a comparison of the experimental values with theoretical predictions based on Eqs. (3) and (4). This comparison was addressed quantitatively by considering the mean of the relative difference between the measured value and the predicted value for the same Froude number. Cases where interaction modes $I$ and $II$ were observed are globally consistent with the energy conservation assumption. For these modes, the mean differences with the prediction using Eq. (3) is 14% for both modes, compared to differences of -20 and -17% for modes $I$ and $II$ respectively considering predictions based on Eq. (4). By contrast, cases where modes $III$ and $IV$ were observed are globally consistent with the momentum assumption. For these modes, the mean relative differences with the prediction using Eq. (4) were -5 and -1% respectively, compared to mean differences of 21 and 40% respectively considering predictions based on Eq. (3). These results clearly show that a significant amount of energy is dissipated in cases where modes $III$ and $IV$ are observed, while the two other modes result in limited dissipation. This is attributed to higher relative displacements between particles during modes $III$ and $IV$ cases, inducing dissipation by friction.

Depth measurements after the flow had stopped revealed that more than 90 % of the tests resulted in a final deposit depth surpassing the initial barrier height. The ratio $h_d/h_0$ between the deposit depth, $h_d$, and the barrier initial height, $h_0$, ranged approximately from 0.9 to 1.6. The fact that the ratio is higher than one is attributed to the coarse nature of the flowing material, with a large ratio of large and angular particles, resulting in interlocking between particles. Besides, due to the barrier top cable lowering, which was observed but not quantified, the ratio between the deposit depth and the effective barrier height is even higher.





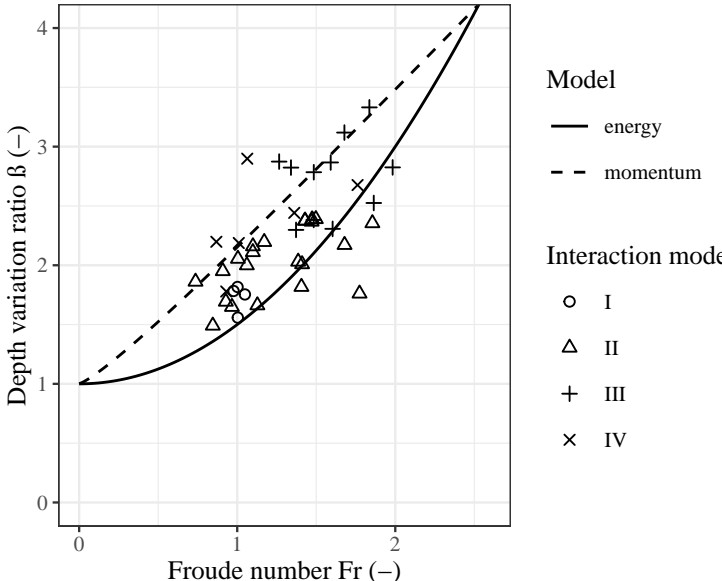

**Figure 9.** Flow depth variation rate as a function of the Froude number. The curves are calculated by the energy and the momentum conservation hypothesis. $I$, $II$, $III$, and $IV$ refer to the interaction modes illustrated in Figure 8.

## 3.3 Barrier response

The barrier response to the loading exerted by the debris flows is first addressed focusing on its deformation. Then, it is addressed in terms of barrier loading, focusing on the situation at rest, in the aim of evaluating the relevance of existing analytical models as for the static component of the loading on the barrier.

### 3.3.1 Barrier deformation

When the flow reaches the barrier, this latter experiences increasing deformation over a 1s duration typically, as illustrated in
Figure 10. This figure also reveals the difference in amplitude and variation with time of the elongation experienced by each cable. The bottom and middle cables are the first to experience elongation, in accordance with the filling dynamics. In this case, the larger elongation was observed at barrier mid-height. This may be attributed to the fact that this cable holds the net above and below it, while the bottom and top cables only hold the net above or below, respectively. The resulting difference in barrier deflection along the vertical axis is illustrated in Fig. 10b. The deformation pattern along the vertical axis was observed to
significantly vary from one test condition to another. In particular, the relative deformation of the top cable with respect to the two others was much more in mode $IV$ cases compared to that in other cases (Fig. 11). This specific feature in the deformation pattern suggests a difference in loading distribution from bottom to top which is considered as a reminiscence of the occurrence of the overflow. In addition, cables globally experience a much larger elongation after tests where mode $IV$ was observed.





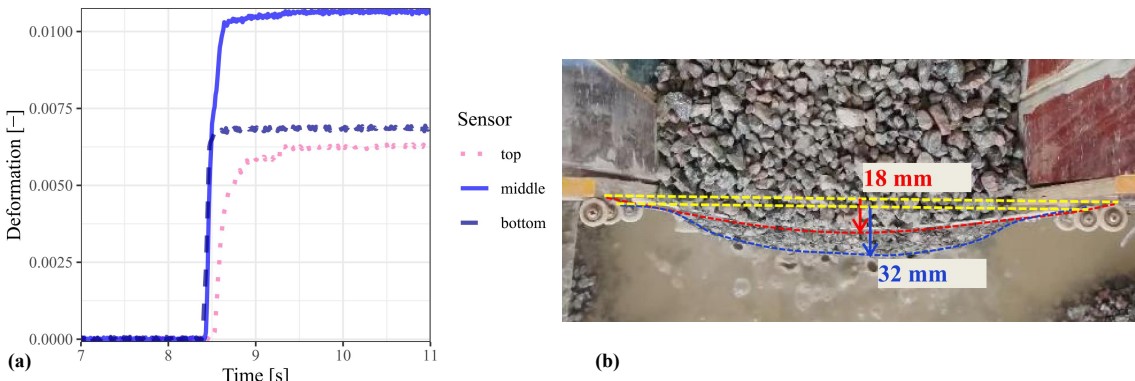

**Figure 10.** Typical barrier deformation: (a) evolution with time of the deformation of each cable and (b) picture from the top showing the barrier deflection (red line shows the top cable and blue line shows the middle cable).

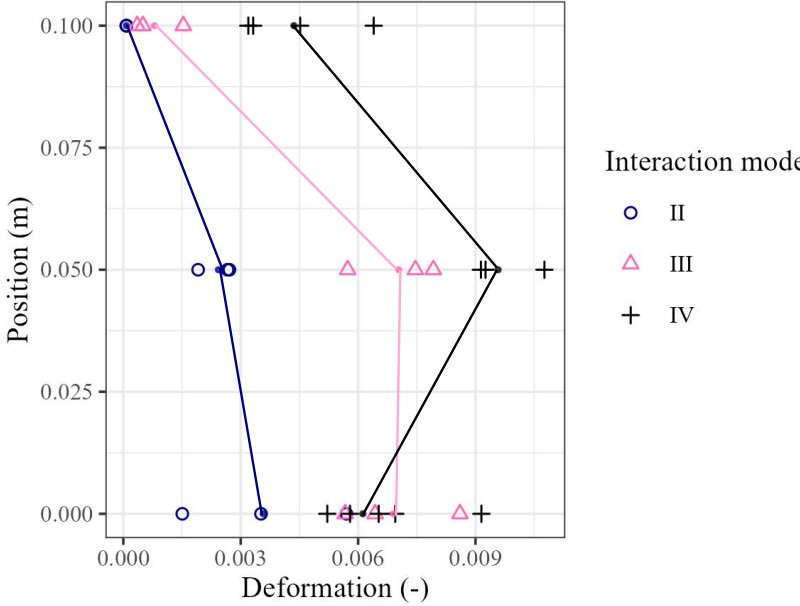

**Figure 11.** Cable vertical position versus relative residual deformation after tests involving cable type 3D2. Dots figure individual tests while the continuous lines figure mean values distinguishing the different modes.



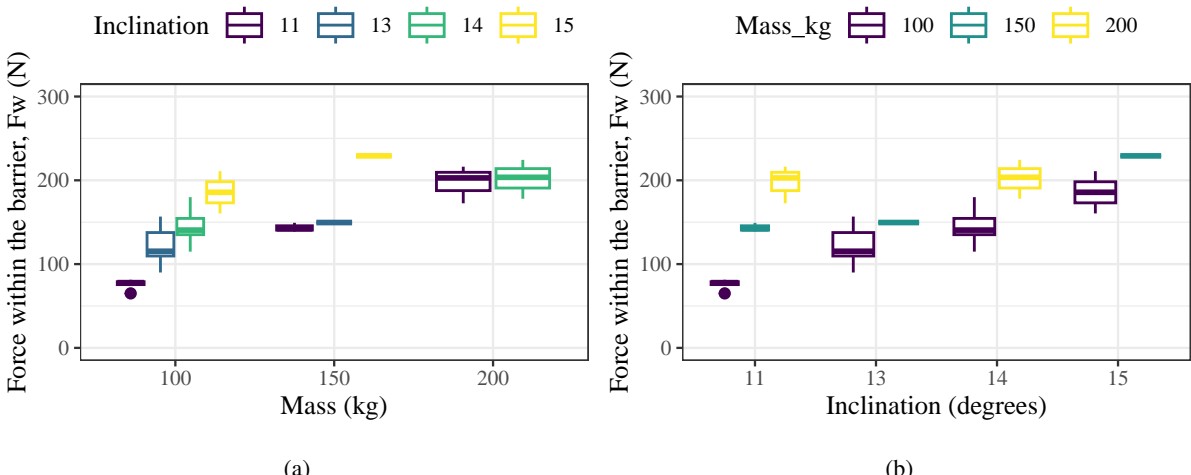

**Figure 12.** Residual force within barrier, $F_w$, as a function of (a) the released mass and (b) the flume inclination.

### 3.3.2 Load within the barrier at rest

The load within the barrier is addressed based on the force measured by the three force sensors, which gives an indication of the amplitude of the force transiting through the barrier towards the barrier anchors. In this aim, the sum of the three forces measured when the system is at rest, $F_w$, is plotted in Fig. 12 showing that the force within the barrier increases with both the inclination and the released mass. In addition, a high dependency of $F_w$ on one parameter is observed when the value of the other is small (i.e. 100 kg and 11 ° for the mass and inclination resp.).

Similarly as for a retaining wall, the loading applied by the retained material on the barrier and consequently the force within the barrier, $F_w$, are expected to be related to the deposit depth $h_d$, which is globally confirmed in Figure 13a. Besides, a global trend where $F_w$ increases with the flume inclination is also observed, even though the variation range of this parameter is small (Fig. 13b). Nevertheless, for a given deposit depth, a ratio exceeding two between extreme values of $F_w$ is observed. This scattering is in part explained by the differences in cable characteristics. Higher $F_w$ values are observed when the net

is supported by pairs of 3D printed cables or by metallic cables. On the contrary, 3D cables lead to lower $F_w$ values at any given depth and any inclination (Fig. 13a.). This is attributed to the fact that stiffer cables restrict the barrier deflection and, consequently, result in higher forces in the supporting cables. This confirms the importance of accounting for similitude in designing barriers to be used in small-scale experiments (Lambert et al., 2023). Last but not least, all cases where mode $IV$ was observed resulted in much higher $F_w$, whatever the cable type. It is particularly notable that for a given deposit depth, a

higher $F_w$ was observed in case the barrier was overflown (Fig. 13a.). This suggests that the residual static loading within the barrier bears a footprint of the dynamic process.





Due to the number of varied parameters, it was not possible to derive analytical expressions reflecting these trends with the tests performed. Nevertheless, these results rather clearly indicate that the static loading exerted by the deposited material, of a given unit mass, is not a function of the deposit depth measured upstream the barrier only.

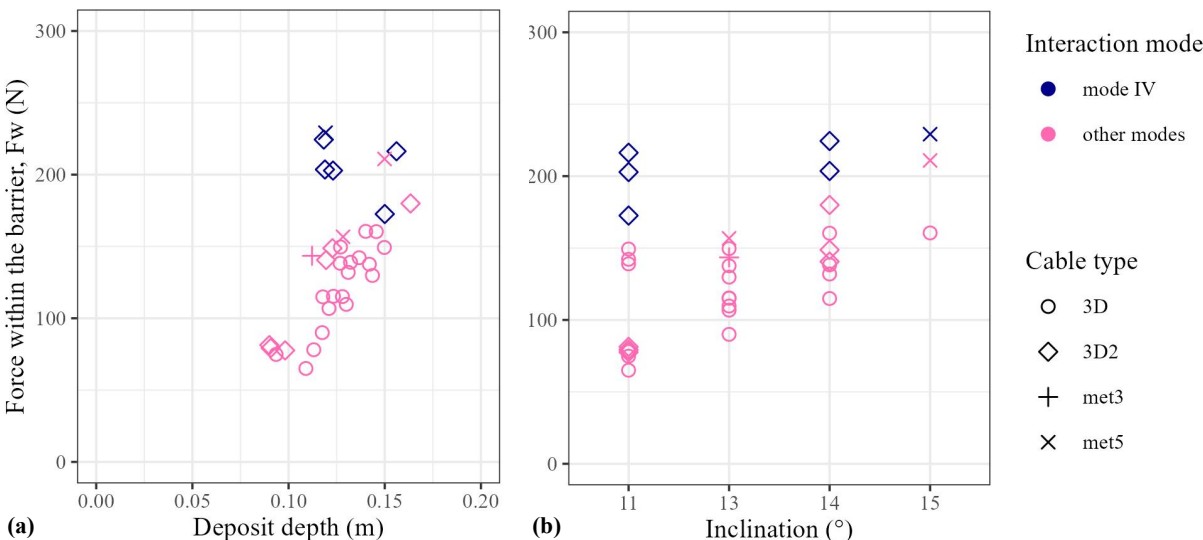

**Figure 13.** Residual force within the barrier, $F_w$, as a function of the deposit depth (a) and as a function of the flume inclination (b).

### 3.3.3 Loading on the barrier

While the residual force within the barrier, $F_w$, is measured transverse to the flow direction, it is possible to compute the loading exerted on the barrier from the elongation and force measurements. The various available analytical models used in this purpose rely on assumptions regarding the distribution and orientation of the loading exerted on the barrier and the barrier deformation along the barrier longitudinal axis (Brighenti et al., 2013a; Ng et al., 2016; Song et al., 2018, 2019; Tan et al., 2019; Lambert et al., 2024). As detailed in Appendix A, the model used for computing the barrier loading assumed an uniform load distribution and considered both the circular and parabolic barrier deformation assumptions. These models were used to compute the total force exerted at rest on the barrier, $F_b$. More precisely, these analytical models were used to estimate the force exerted on each cable and $F_b$ was computed as the sum of these three forces.

The force acting on the barrier when at rest was also computed based on analytical expressions from the literature dealing with the design of flexible barriers exposed to debris flows. This static force, $F_s$, is classically computed as follows:

$$F_s = \frac{1}{2} K \rho g h^{*2} B \tag{5}$$

where $K$ is the lateral pressure coefficient, which is is generally set to 1.0 assimilating the loading to the hydrostatic case, as in Berger et al. (2021). $\rho$ is the deposited material unit mass, which was set to 2200 kg/m$^3$ from the measurement on the initial





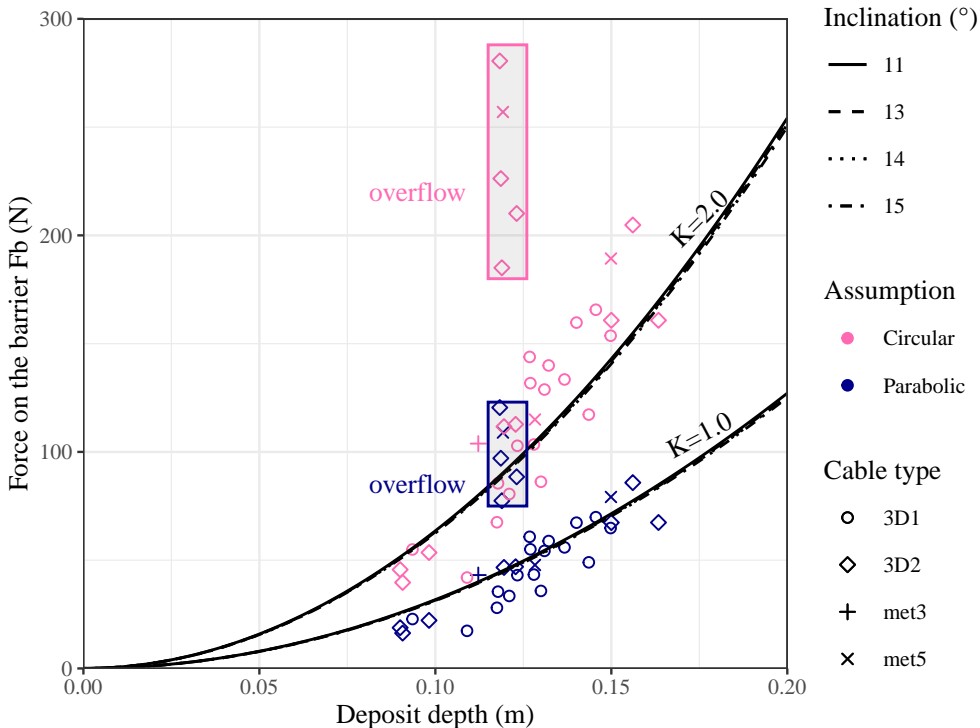

**Figure 14.** Residual force applied on the barrier, $F_b$, as a function of the deposit depth. The dots in the pink and dark blue boxes refer to overflow cases (i.e. interaction mode $IV$).

mixture. $B$ is the barrier length. $h^*$ is the depth of material upstream the barrier. In design practices, $h^*$ is the depth of material
at rest close to the barrier and never exceeds the barrier initial height. This depth was not measured during the experiments. In
lieu, $h^*$ was considered equal to the deposit depth, $h_d$, which was measured 10 cm from the barrier, keeping in mind that this
value was generally higher than the depth close to the barrier and often revealed higher than the barrier initial height by a ratio
of up to 1.6.

The first general observation is that the circular shape assumption for the deformed barrier results in a total force $F_b$ as
much as three times higher than when considering the parabolic assumption. This observation justified evaluating the validity
of each shape assumption. In this purpose, the barrier deflection predicted by each model was compared to that measured for
the specific case shown in Fig. 10b. The parabolic assumption resulted in deflection values of top and middle cables of 14 and
20 mm, respectively. Values of 36  mm and 52  mm were respectively obtained considering the circular shape assumption.
Comparison with the measured values (18 and 32 mm, respectively) reveals that the parabolic assumption is more appropriate.
Based on this, the parabolic assumption was considered the most relevant hypothesis. Since both hypotheses result in very
different values of impact force (and thus strong differences in the design and cost of structures), this is a key result of the
present work which is less subject to scaling issues than previous works because it is the first to address this process using



debris flow material made of mixture of gravel, sand, clay and water, flowing in a realistic $Fr$ range, and using a small scale model of the flexible barrier which was design accounting for mechanical similitude requirements.

The second general observation is that, for a given deposit depth, $F_b$ is variable from one test to the other, which confirms that the deposit depth is not the only parameter with significant influence on the loading on the barrier at rest. The residual force in overflow cases (black frames) is globally much higher than that in the cases where all the material is retained with average and maximum ratio between extreme values of about two and four respectively. Since debris flows very often present several surges and flexible barriers have only a limited capacity, most of them should be expected to overflow in due time. This

difference in load between the two situations is thus also important to account for in barrier design recommendations.

        The third general observation is that, without overflowing, all relevant points (i.e. using the parabolic assumption) globally align with the prediction based on the hydrostatic loading model, i.e based on the analytical model expressed in Eq. (5) while considering a pressure coefficient of 1.0. Meanwhile, the overflowing cases result in a total force $F_b$ closer from predictions considering a pressure coefficient of 2.0.

## 400  4   Discussion

In this research, a particular focus was placed on the deformation and loading of a flexible barrier at rest, after it intercepted a debris flow. The motivation for this was that the static component of the force exerted by the deposited material (or dead zone) on the barrier received limited attention up to now, while it may have a major contribution in the barrier loading. This is in particular the case when the flow Froude number is in the range of events observed in Alpine environments. The postulate

behind this is that the state at rest can be assimilated to the force exerted on the barrier, during an event, when the deposited material has the same depth. This situation may correspond to a given time instant during the progressive debris accumulation or to the case of a surge reaching the barrier which already caught debris material. Because in this study the ratio between the maximum deposit depth and the barrier initial height, $h_d/h_0$, ranged approximately from 0.9 to 1.6, these results rather concern the second situation with a filled or almost filled barrier.

This ratio ranging up to 1.6 can also be read in another way: the surge depth varying in the 50–100 mm range (Fig. 6), the total deposit in case of filling without massive overtopping can be computed as the sum of the barrier height plus the depth of an approaching surge that would stop right before the barrier. In our test campaign, the maximum deposit height is indeed about the net height plus the average of the surge height between every tests. Considering the maximum surge height would be too conservative because the biggest surges are also more mobile and thus less likely to stop right at the barrier. Consequently,

it is believed that this large ratio refers to this barrier height, and that it would reduce increasing the barrier height.

        It was shown that, in absence of overflow, the total force applied at rest on the barrier $F_b$ could be reasonably modelled assimilating it as an hydrostatic loading for any deposit depth. Nevertheless, this conclusion was reached considering the maximum depth of the deposit, and not the depth close to the barrier as generally done (Berger et al., 2024). According to this latter approach, the deposit depth for 90% of the tests should be restricted to the barrier height (10 mm), leading to the situation

where all the points above 100 mm are shifted to align parallel to the y-axis in Fig. 14. Of course, this would not be relevant



because for a same depth (100 mm) the force would vary over a large range, with a ratio of four between extreme values. On the contrary, these results suggest that the relevant depth to consider in our case where the deposit depth is not uniform should be the maximum depth, which was most often observed at distance from the barrier.

The use of Eq. (5) requires determining a pressure coefficient. A pressure coefficient value of 1.0 revealed appropriate for computing the total force at rest, $F_b$, in case there was no overflow. This value is consistent with that used by various authors, such as Wendeler et al. (2019); Berger et al. (2021). In the case where overflow occurred, Fig. 14 suggests that $F_b$ could be reasonably well captured considering a value of 2.0 for this coefficient in our experimental set-up. It is important noting that using Eq. (5) implies a triangular stress distribution along the barrier height, decreasing to zero from barrier bottom to top. This hypothesis is not appropriate in this case. Indeed, results presented in Fig. 11 suggest that the upper part of the barrier experienced a significant load in case overflow occurred, which is consistent with previous research results (Wang et al., 2022). In brief, Eq. (5) was used for convenience and for comparison purpose, but not for suggesting any overflow-induced load distribution along the barrier height.

No reverse barrier deformation was observed after the flow stopped, even when overflow was observed. This latter observation reveals that the granular body matrix keeps the memory of the overflow and supports the hypothesis that the state at rest can be related to the static loading exerted during the event by the deposited material on the barrier. The much higher values of $F_b$ in case overflow was observed reveals that the barrier is exposed to a flow-induced loading which is transient and vanishes to zero when the flow stops.

The large difference in total force exerted at-rest on the barrier between the two situations is first considered as a consequence of the surcharge, $q$, on the deposited material associated with the flow depth above it. The additional barrier loading induced by this surcharge, $F_{bq}$, was estimated considering the Rankine's theory of earth pressure and assimilating the flexible barrier as a vertical rigid wall and neglecting the flume inclination. An active earth pressure coefficient in the 0.3-0.55 range was considered to account for the uncertainties associated with the material friction angle and surface inclination. The maximum flow depth during overflow being 65 mm, the total force acting on the barrier due to this surcharge, $F_{bq}$, was estimated to be in the 13-23 N range. Besides, predictions based on Eq. (5) and considering the typical deposit depth measured after overflow occurred (0.12 m approx) give values of 47 and 94 N for the hydrostatic force, $F_{hydro}$, and the total force, $F_b$, acting on the barrier respectively. The total force component attributed to overflow ($F_b - F_{hydro}$) thus equals 47 N. This result indicates that the surcharge $F_{bq}$ is responsible for only 27–49% of the force component attributed to overflow.

The remaining force ($F_b - F_{hydro} - F_{bq}$) amounts 24-34 N. It is attributed to the shear between flowing and non-flowing particles that develops at the interface between the flow and the deposit (Albaba et al., 2017; Wang et al., 2022). This remaining force, which is referred to as $F_{bs}$, appears not negligible in this case, which contrast with suggestion by Berger et al. (2021).

These results globally evidence rather clearly that the barrier overload when overflow occurs accounts for both a gravity load and a shear induced load, which precise quantification requires further investigations.



**Conclusion**

This paper presents small scale experiments of flexible barriers impacted by near-critical debris flows (i.e. $Fr$=0.9–2). These
tests were carefully designed to model quite realistic debris flows with a mixture of gravel, sand, clay and water hitting flexible
barriers manufactured with 3D printers such that their flexibility is in mechanical similitude with actual steel barriers. The
impact regime and deformation of both the stopping debris-flow material and the barrier are modelled with less scale effects
than in previous works using simpler mixtures (e.g. dry sand), excessively fast flows (i.e. $Fr \gg 1$) or with irrelevantly-stiff
barriers (e.g. made of steel or of nylon).

Focusing on a narrower range of $Fr$ and ensuring a relevant deformation of the flexible barrier enabled to highlight a
gradual change with the flow kinematics in the way the debris flow material is stopped. Four modes are described from a
mass immobilisation with very limited material reorganisation for low flow energy ($Fr \approx 1$) to high granular jump leading to
material accumulation when flow kinetic energy increases. The greater the granular debris flow material reorganizes through
this piling-up, the greater the dissipation of energy by friction within the material. The analysis of the forces and deformation
within the barrier demonstrate that between the two existing deformation models, namely the circular and the parabolic, the
latter is the most consistent with the measurements. We could then verify that the static loading exerted on the barrier can be
predicted based on an hydrostatic pressure model when the maximum deposit depth is taken into account.

For a sufficiently high accumulation, the flexible barrier is eventually overtopped by the flow and only part of the flowing
material is trapped. Interestingly, overflow resulted in a significant increase in the residual load on the barrier upper part, and
in an equivalent static force acting on the barrier typically twice that observed in the absence of overflow. We interpret this
doubling of the force to be due to the surcharge associated with the flowing material, which depth is significant compared to
the barrier height in our set-up, and to a flow-induced shear at the surface of the trapped material. Considering that flexible
barriers have a limited trapping capacity and that debris flows usually occur in series of surges, this additional loading deserves
more attention in future researches as it might be more important in the design than the usually-studied single surge impact.

*Code and data availability.* The code used to process the data and the sensor data are available upon reasonable requests.

*Video supplement.* Videos of some experiments can be downloaded on https://cloud.univ-grenoble-alpes.fr/s/oWj2a72LPj4RdY2 and will
be uploaded on the repository https://entrepot.recherche.data.gouv.fr/dataverse/filtor at the revision stage of this paper.





**Appendix A: Computation of the load applied on the barrier**

Retrieving the force acting on the barrier requires defining a model relying on hypothesis concerning, first, the distribution of

the flow-induced load to account for and, second, the barrier deformation (among others, see Brighenti et al., 2013a; Ng et al.,
2016; Song et al., 2019; Lambert et al., 2024).

As for the load hypothesis, a general consensus consists in considering a uniform distribution along the barrier length. By
comparison with other distributions (triangular or parabolic), this distribution was in particular considered more appropriate
for the static load estimation (Wendeler, 2008b; Wendeler et al., 2018). As for the second hypothesis, the barrier deformation

along the channel transverse axis is either considered circular or parabolic. In the first case, the loading is considered normal
to the deformed barrier (Song et al., 2018, 2019) while in the second case, it is considered parallel to the channel direction
(Brighenti et al., 2013a; Wendeler et al., 2018; Huo et al., 2023). Both these shapes were considered in this study and their
results were compared.

From the pattern of the parabolic curve depicted in Fig. A1a, at a certain time of the debris-flow impact, the deflection of a

given cable in the flow direction y along the width $u(x)$ is expressed as follows:

$$u(x) = \frac{q}{2T_x}(x \cdot l - x^2) \tag{A1}$$

where $q$ is the uniformly distributed load (N/m) along the initial length of the cable $l$ and $T_x$ is the component of the tensile
force $T$ along the x-axis. The maximum deflection $u_{max}$ is observed at the centre point of the cable and is given by:

$$u_{max} = u(x = \frac{l}{2}) = \frac{ql^2}{8T_x} \tag{A2}$$

From the elongation of one cable $\Delta l$ and ignoring the higher differentiation, the maximum deflection $u_{max}$ can be calculated
using:

$$L - l = \Delta l = \frac{8u_{max}^2}{3l} \tag{A3}$$

where $L$ is the length of cable once deformed with a parabolic shape. Therefore, the deflection angle at one cable's extremity
$\theta$ is obtained by solving the derivative of the curve along the width at the cable's extremity:

$$\tan \theta = \frac{du}{dx_{|x=0}} = \frac{ql}{2T_x} = \frac{4u_{max}}{l} \tag{A4}$$

Combining equations A4, A2 and A3, it comes:

$$\tan \theta = \frac{2}{l}\sqrt{\frac{3l \cdot \Delta l}{8}} \tag{A5}$$

In total the normal load acting along the cable is solved by:

$$F_n = 2T_y = 4T_x \sin\frac{\theta}{2}\cos\frac{\theta}{2} = 2T_x \cdot \sin\theta \tag{A6}$$





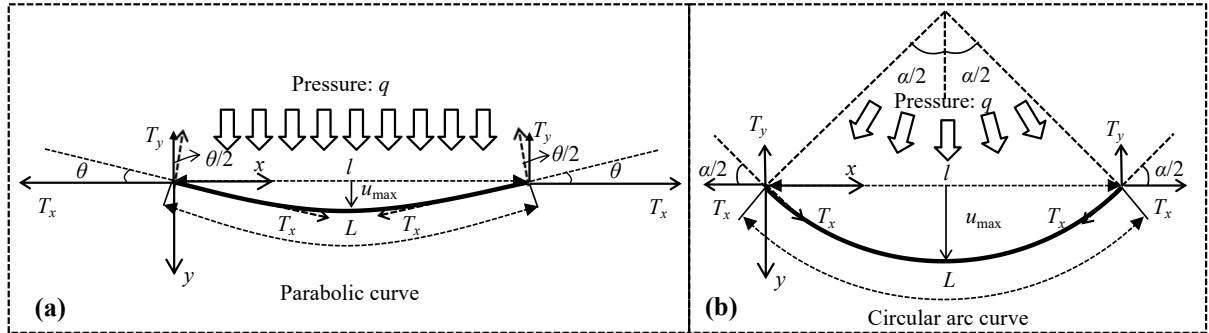

**Figure A1.** Deformation assumption of the cable subjected to normal debris-flow loading. Note that the calculation of the tensile force here is different from that in (Brighenti et al., 2013b; Song et al., 2018, 2019) due to the setting of pulleys at the extremities of the cable.

As such $F_n$ is back-calculated depending on the tensile force measurement and the elongation $\Delta l$. It is noteworthy that in this study the barrier cables are deviated by pulleys in such a manner that $T_x$ is measured by the force sensors.

For the circular assumption (Fig. A1b), the impact load is uniformly perpendicular to the deformed cable. According to Song et al. (2019), the form finding of the cable is explicitly based on the curvature angle of the circular arc $\alpha_i$ and the arc length $L$.

$\alpha$ can be related to the initial and arc length of the cable by the Taylor expansion as Sasiharan et al. (2006):

$$\frac{\alpha}{2} = \sqrt{6(1 - \frac{l}{L})} \tag{A7}$$

Here the deflection angle $\theta$ is equal to $\alpha/2$, yielding the same calculation expressed by eq. A6. It is noteworthy that only the component of the total load pressure from the flow direction that matters because the component of the total load pressure from the direction that is normal to the flow direction is counterbalanced. And the maximum deflection can be calculated by:

$$u_{max} = \frac{0.5l}{\sin(\alpha/2)}[1 - \cos(\alpha/2)] \tag{A8}$$

*Author contributions.* HM: funding acquisition, resources, supervision of experiments; HM & GP: software, data curation, formal analysis; SL, HM & GP: conceptualization, investigation, visualisation, writing - original draft preparation; FF prepared the depth and elongation acquisition system.

*Competing interests.* The authors declare having no competing interest regarding the results of this paper.



*Acknowledgements.* This study was funded by the Natural Science Foundation of Sichuan Province (2022NSFSC1123) and China Scholarship Council (CSC). All the tests were performed by a group of master students from Sichuan Agricultural University, China on an experimental set up designed and co-equipped together with INRAE.





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
