# Peer review of "Capture of near-critical debris flows by flexible barriers : an experimental investigation"

_EGUsphere, 2024_

## Author Comment (AC3)

Title: Capture of near-critical debris flows by flexible barriers: an experimental investigation

Author(s): Miao Huo et al.

MS No.: egusphere-2024-3575

MS type: Research article

Third referee comments and replies   -(June 13th 2025)

**General comments**

The authors carry out flume experiments focusing on replicating low Fr conditions in Alpine region. The objective was to identify different impact modes on a flexible barrier. I would like the authors to improve the overall story of the manuscript by explaining why static force and effects of overflow are important for barrier design in Alpine region in the literature review. Currently, the authors superficially criticise the existing literature on barrier impact for focusing on dynamic loading, which was done so to suit the requirements in mountainous regions such as Hong Kong. If the author wants to claim novelty in terms of identifying 4 modes of flow-barrier interaction, I suggest including plates for different time steps for each mode to help explain the distinctions. Some phrases are overly complex or lengthy, obscuring the main points. Simplifying sentences and removing unnecessary jargon can enhance clarity. I would like to recommend major revision for this manuscript.

> Authors' response:
> The authors thank the referee for the comments made. All the general and specific comments have been thoroughly considered. Most of these have resulted in changes to the manuscript as described below.

**Specific comments**

1.    Abstract:  "Four different interaction modes between the flow and the barrier are identified increasing the flow kinematics". Can you be specific why these so called "interaction modes" are important in terms of mechanisms and in terms of application? What is meant by "increasing flow kinematics"?

> Authors' response:
> A very large number of academics have addressed the impact force of a debris flows on an obstacle. Various model for the impact force have been proposed, making a distinction depending on the interaction modes (e.g. pile-up or run-up). Indeed, the developed models are based on assumptions that basically depend on the impact mode (e.g. Sun and Law, 2015; Ashwood and Hungr, 2016; Poudyal et al., 2019). First, our experiments revealed no run-up mode for low Froude number flows consisting in coarse and high solid fraction material. In addition, it was observed that one mode couldn't be assimilated to the pile-up mode and was never described before (mode I, for lower Fr). As for mode II, it vaguely resembled the pile-up mode. Mode III could be assimilated to the pile-up mode. All this motivated the authors in describing these different modes, to give a better picture of what the authors observed (and which slightly differ from previous descriptions). The differences between these modes is detailed in the text, with sufficient details. Last, modes I to III were observed for increasing Froude numbers, meaning for "increasing flow kinematics".

References
- Sun, H. and Law, R.: A preliminary study on impact of landslide debris on flexible barriers, Tech. Rep. 309, Geotechnical Engineering Office Civil Engineering and Development Department The Government of the Hong Kong Special Administrative Region, https://www.cedd.gov.hk/filemanager/eng/content_557/er309links.pdf
- Ashwood, W. and Hungr, O.: Estimating total resisting force in flexible barrier impacted by a granular avalanche using physical and numerical modeling, Canadian Geotechnical Journal, 53, 1700–1717, https://doi.org/10.1139/cgj-2015-048, 2016.
- Poudyal, S., Choi, C., Song, D., Zhou, G., Yune, C., Cui, Y., Leonardi, A., Busslinger, M., Wendeler, C., Piton, G., and Moase, E.: Review of the mechanisms of debris-flow impact against barriers, in: International Conference on Debris-Flow Hazards Mitigation: Mechanics, Prediction, and Assessment, p. 1027–1033, https://doi.org/10.25676/11124/173112, 2019.

2.     Cases with higher Froude numbers can be found in Hong Kong where the requirement for the design of barriers is their precarious location of immediately downstream of steep terrains. In contrast, the requirement for Alpine region is different. Therefore, the authors should not be too bold to criticise that none of the barrier design guidelines consider lower Fr. The literature review section only discusses how other studies did not consider low Fr numbers for the barrier design. In my opinion that is not sufficient for novelty. Explain the significance of considering the low Fr dynamics. Also provide an extensive review on previous literature that modelled down-scaled flexible barriers while identifying their limitations.

Authors' response:
The authors fully agree with the referee that high Froude number exist in nature, and in particular in steep gullies. This is in particular the case upstream elements at risk such as the city of Hong Kong for which it is recommended to consider Fr>3 (The referee will note that Kwan et al (2013) described a debris flows in Jordan Valley, Hong Kong, with Fr < 2). Research aiming at protecting the city of Hong Kong is very active since the beginning of the 2000's. This explains why the vast majority of technical documents and articles on debris flows (in particular from Asia) deal with high Froude number flows. Also, academics have highlighted the difficulty in conducting small-scale experiments in flumes for dealing with low Froude numbers flows. As a consequence, the developed state of the art mainly relates to flows with a Froude number higher than 2.5, which do not correspond to Alpine environments contexts. This motivated the authors in writing: "As yet suggested by Hübl et al. (2009), it can be considered that models in relation to the flow-barrier interaction have globally been developed on an input data range which does not comply with the most frequently observed debris flows type in Alpine areas." Latter, one can read "*In the end, it appears that most of the published findings which serve as basis for improving design methods of mitigation structures concern supercritical to highly supercritical flows*.". In fact, what the authors say is that most of the recent research aimed at improving design guidelines does not concern debris flows with a Froude number of

less than 2. Doing so, the authors highlight the need to develop knowledge related to low Froude numbers flows, in the continuation of some well known research such as that from Ashwood and Hungr for example.

As for the significance of considering low Fr dynamics, one can read in the introduction of the submitted manuscript "… *This difference in Froude number results in a difference in loading regime on the barrier. Indeed, many authors have evidenced that above a Froude number of ≈1.4, the loading exerted by a granular flow onto an obstacle was dominated by inertia forces, which relate to the flow velocity, and that, below this value, the loading on the barrier was mainly dominated by gravity forces, associated with the depth of the intercepted material (Tiberghien et al., 2007; Laigle and Labbé, 2017; Wendeler et al., 2019; Huang and Zhang, 2022). In other words, the difference in Fr between most research conditions and what is observed in nature leads to an excess in the attention on the influence of the flow velocity, which has consequences on the knowledge as for the way the flow accumulates, deposits and overflows the structure and more generally interacts with it.*" The authors believe that this provides the reader with sufficient basic explanation  and some references for supporting our motivation in dealing with debris flows observed in Alpine environments.

It is true that the response of certain flexible barriers to flows (dry granular to muddy debris) has been addressed by some scholars based on small-scale experiments. The limitations with these studies are mainly related to the limited attention paid to similitude requirements. This motivated the research presented in Lambert et al. 2023, where the limitations with these previous works are mentioned.  In addition, most of these studies addressed high Froude number flows and focused on the peak load. By contrast, the proposed manuscript concerns near critical flows and mainly focus on the loading at rest. For these reasons, it appears not necessary going in a presentation of all previous published studies involving small scale flexible barriers.

References
-   Kwan, J.S.H., Chan, S.L., Cheuk, J.C.Y. et al. A case study on an open hillside landslide impacting on a flexible rockfall barrier at Jordan Valley, Hong Kong. Landslides 11, 1037–1050 (2014). https://doi.org/10.1007/s10346-013-0461-x

3.    Para 60: "There is thus a vital need for an in-depth investigation of the flow-barrier interaction while considering debris flows with a Froude number closer to that in most frequent real cases, i.e. <2, in particular in view of improving the design of mitigation structures such as flexible barriers." Here you may want to highlight the objective is to consider "cases in the Alpine region."

Authors' response:
Correct, this sentence was improved.

4. Para 75: explain the basis behind selecting the varying parameters as the flume inclination and the flow mass. What was the reason to consider different barriers? Introduce in text what is the scientific difference considered.

Authors' response:

The mass and flume inclination were varied in view of covering the range of Froude numbers observed in Alpine environments while considering the same material (which is a granular debris flows with high solid fraction). An other strategy for varying the Froude number could have consisted in varying the flowing material characteristics. This is a different approach, with a different aim. The approach considered by the authors may be analogous as some kind of site specific study where the flowing material characteristics are well known, but the volume of the surge and the location of the barrier (and thus torrent bed inclination) are not known a priori.

As for the barrier, the parameters concerned the number of cables and the type of cable used as horizontal cables. There was no other difference. The manuscript was revised to explain why barriers where equipped with three 3 D printed horizontal cables, three pairs of 3 D printed horizontal cables or three metallic horizontal cables. Basically, the main reason was that barrier partial or total failure was observed when a barrier supported by three 3 D printed horizontal cables was exposed to a surge with a high mass or high velocity. The results discussion was also improved to address the influence of the barrier design on the loading within the barrier.

5. Pg 4, Para 95: Provide citation for the mixture composition of the "real debris flow" from field cases in Alpine region.

Authors' response:
The criteria we followed and our references are now clarified as follow: "The characteristics of the mixture (wide grain size distribution, clay into the interstitial fluid, water content <50%) were determined to mimic coarse-grained debris flow with a high solid fraction commonly found in Alpine region (Coussot and Meunier, 1996; Hürlimann et al., 2003; Brenna et al., 2020)."

6. Pg 5. Para 120: What is a "water drop mesh net"? Review the "downscaled flexible barrier" used in Ng et al (2016) and Wang et al. (2022) (Wang, L., Song, D., Zhou, G. G., Chen, X. Q., Xu, M., Choi, C. E., & Peng, P. (2022). Debris flow overflowing flexible barrier: physical process and drag load characteristics. Landslides, 19(8), 1881-1896).

Authors' response:
In fact, line 120 reads "a net with a water-drop mesh net" which means that the net consists of the repetition of a unit mesh having the shape of a water drop (see image below). This net is commonly used in France and was used by the authors in the previous research they cite (Piton et al, 2023; Lambert et al, 2023-2024). The beginning of this section was rephrased to better describe the net and refer to previous research.

[Figure]

[Figure]

Drawing and picture of a water-drop mesh (from Lambert et al. 2023)

Ng et al (2016) considered a continuous and flexible plastic sheet in lieu of the barrier and horizontal cables equipped with energy dissipators. Similitude was clearly addressed designing the dissipators only. Wang et al (2022) considered a net made from the repetition of a rhomboidal mesh, withouot giving any explanation on the barrier down-scaling principles. In both these studies, energy dissipators were considered which is not the case in the presented study. Energy dissipators have a huge influence on the barrier response, in terms of load on the anchors in particular, as evidenced in Albaba et al (2017). This influence is much higher than that of the mechanical characteristics of the net, because it results in a tremendous increase in cable lenght.

7.  Pg 6, Para 125: The description is not clear about the horizontal and vertical cables. Are all horizontal and vertical cables made from the S-resin? What do you mean "single and pairs of cables" were used in tests? Does that mean the used number of cables are not consistent between tests? Later, authors also state that "metallic cables" were used but it is not clear where. Why are two materials used for cables? Figure 3 must be improved to include all the main cables with the net including a proper naming convention for cables. In Fig 3(b), please include annotations for lengths and diameters. What is the unit of dimensions shown (Fig 3(a))? According to Table 3, varying number of cables and material are used for the tests. In order to avoid confusion, I suggest the authors clearly explain the basis behind different cable selection in the Paragraph 125.

Authors' response:
The text and figure describing the net and its components have been completely rewritten.

8.  Pg 8, Para 175: If the measurement of deposit height at the point upstream of the barrier (US2) is uncertain, how can you use this data to reliably interpret the test results?

Authors' response:
In fact, side videos allowed circumventing the limitations with US2. To be more explicit, measuring automatically the deposit depth when the surface of this deposit was not parallel to the flume bottom resulted in doubtful values. This is the main

reason why, the authors decided to use the side video images for obtaining the maximum deposit depth. The referee will note that, due to the maximum grain size, the surface of the deposit is not smooth, which reduces the precision in the depth measurement. However, the differences in deposit depth observed between all test series is much larger than the uncertainty of this measurement, which allowed the authors using this data in the results interpretation. The text describing the data post treatment was improved to better present this.

9. Pg 12, Para 245: Clearly explain the difference between mode-I and the layering mechanism observed for frictional flow. What is the reason for the "vertical expansion"?

Authors' response:
In fact, mode 1 is closely related to the grain size and grain shape, in addition, to the Froude number of the flow for which mode 1 is observed. The maximum grain size is 20 mm while the minimum flow depth is 50 mm and the barrier height is 100 mm. In addition, grains have irregular shapes. It is important reminding this before describing what happens when the flow (with Fr very close to 1) is arrested by the barrier. The first comment is that, as intuitively expected, the front material stopped first when reaching the barrier, but the rest of the material stopped moving in a very short period of time after the front touched the barrier. The flow depth, which was rather uniform, increased uniformly during this braking period, in such a manner that the deposit surface resulted parallel to the flume bottom. The ratio between the deposit depth and the flow depth was 1.2 on the average. Typically, the incoming flow depth corresponded to 4 times the maximum grain size and the deposit depth corresponded to 5 times the maximum grain size. The authors believe that the observed vertical expansion results from the squeezing of the arrested material by the moving material. In other words, once a given volume of material is arrested, it experiences a stress by the moving material which results in a decrease in its length (along the flume longitudinal axis) and an increase in its depth (perpendicular to the flume bottom). This implies grain compaction and rearrangement, without any piling up. This is the reason why the authors referred to this mode as the quasi-solid body behaviour mode. The text was improved to better explain this.

10. Fig 8(b): Please mark the amplitude of the reflected wave (mode-II). Demarcate the interface between the incoming flow and deposited material.

Authors' response:
In fact, some images correspond to situations at rest (a, b), while others correspond to time instant during the flow (c, e, and f) . This was made explicit in the new version of the figure. When relevant, the interface between the incoming flow was shown.

11.	Pg 13, Para 270: How can an accumulated volume with a depth twice the height of barrier be arrested behind the barrier? Explain the physical mechanism that stabilises the arrested volume.

Authors' response:

This was briefly described in line section 3.2.2 (lines 319-320). This part was extended based on the following.

The first comment is that having a maximum deposit depth higher than the barrier initial height was a general trend. We would like to stress that debris flow deposit above check dam crests are very common in torrentsof the French Alps for instance. Also, in this study lower deposit depths were globally observed for lower released mass or lower flume inclination (mainly associated with interaction Type 1).

The considered mixture contains a rather high percentage of coarse and angular particles which characteristics explain the high deposit depth. There is a progressive increase in depth of the deposit starting from the barrier. Due to friction with the deposit and to a lower angle for the flow to propagate, the incoming material decelerates and progressively comes at rest while flowing above the deposit before reaching the barrier. In this process and if the released volume is large, the value of the maximum deposit depth progressively increases over time and the location where the maximum depth is observed shifts upstream. In the end, the deposit surface is not parallel to the flume bottom but exhibits a peak at a certain distance from the barrier. The deposit surface thus shows a downstream-oriented slope upstream the barrier with an angle less than the angle of repose of the material. The shape of the deposit, and the fact the maximum depth of the deposit can be as high as twice the barrier height (without any significant material release) is clearly attributed to the size and angularity of the grains in the mixture, which in particular control friction with the deposit and angle of deposit surface upstream the barrier.

However, if the released material is clearly in excess with the barrier capacity, the material which is at rest above the barrier height is flushed away by the incoming flowing material. In such a case, the final deposit surface is almost parallel to the flume bottom.

12.	Pg 15, Para 310-315: It is not accurate to simply say "globally consistent" when comparing the theoretical line and experimental data for modes. Explain why some points from mode III and IV cannot be conservatively predicted even by the momentum equation. What is the basis of stating points with higher beta values relate to higher energy dissipation? Explain in manuscript clearly.

Authors' response:
The intent with this section was to compare the obtained results with prediction from two well known theoretical models to see if some conclusions could be derived, in particular in terms of energy dissipation depending on the interaction mode. Even though these models were established based on sound theoretical approaches, the fact is that they are based on various assumptions (hypotheses), concerning the flowing

material and its characteristics. Among other assumptions, it is considered that the flowing material is incompressible and homogeneous or that the change in flow depth is abrupt. These hypotheses are not valid in the treated case. The estimation of the Froude number has also some uncertainties. All of these reasons are thought to explain why some experimental results do not align with the predictions.

Another reason is that the different modes are defined based on visual observation. The classification of a given test between mode II or mode III is thus not based on an undisputable approach. This results presentation may thus be biased by some classification errors. This is the reason why the authors discussed this figure in terms of general trends, without focusing on particular values.

However, the text was modified to be less affirmative with the conclusion and to better explain why it is suggested that modes I and II are associated with lower energy dissipation (and conversely for the other modes).

13. Pg 19, Para 365: Number the equations considered with circular and parabolic assumption both in the text and in Fig 13 legend.

   Authors' response:
   This comment was accounted for (text and Fig. modified)

14. The terms "3.3.2 load within the barrier" and "residual force within barrier" both seem to refer to F_w. Please be consistent and stick with one term to avoid confusion. Later on, 3.3.3 "loading on barrier" further adds to the confusion. You may explicitly use "measured residual force" and "calculated residual force". What's the difference between Fw, Fs and Fb? Which figure are you talking about in section 3.3.3? it is not clear if Fig 14 has measured Fb values or calculated.

   Authors' response:
   The text was revised to avoid any ambiguity (the fact is that there was one incorrect sentence). In brief, the load within the barrier (Fw) is the sum of the forces measured at the extremity of the three cables after the material is at rest and it is addressed in section 3.3.2 and fig. 13. The loading on the barrier (Fb), also at rest, is computed based on appendix A, and it is addressed in section 3.3.3 and illustrated in fig. 14.

15. Pg 20: In addition to verifying that parabolic assumption is appropriate using calculated force, can you use the measured peak deflections of cables to provide evidence?

   Authors' response:
   As evidenced in Figure 10, the elongation of the cables experienced no significant variation after reaching its peak value. Besides, no images were taken during the event to measure the barrier deflection. Last, the peak value of the force was not precise enough for being used in this study. As a consequence, the authors regret they can't go further here.

16.  Estimate the drag load using the method in Ng et al. (2022) (Ng, C. W. W., Majeed, U., & Choi, C. E. (2022). Effects of solid fraction of saturated granular flows on overflow and landing mechanisms of rigid barriers. Géotechnique, 74(1), 27-41.) As explained by the authors the overflow occurs after a certain amount of material is arrested by the flexible barrier. Does that mean a dynamic peak can be observed after an initial static force in the force evolution? Show the force evolution and explain this observation.

Authors' response:
As explained in section 2.5, the force sensors were mainly used to determine the force at rest, after the material had stopped flowing. The measurement of force with time was considered not reliable for being accounted for. The authors thus focused on the barrier loading at res and it is not possible to reply positively the last question and suggestion.

The effect on the barrier of a flow on top of the dead zone is complex to assess. In case of overflow, two different components are generally considered: a normal stress on the dead zone surface and a shear stress at the interface between the flowing material and the dead zone (=drag). This is illustrated in the image below, taken from Berger et al, 2021. By contrast with the former (which requires the flow depth and unit mass only), the computation of the latter faces some questions related, in particular, to the friction angle at the interface between the flow and the deposit (see Kwan, 2012, which is cited in Ng et al., 2022). More importantly, the way these two contributions transfer to the barrier lies on assumptions and requires defining some models. There is a general agreement for computing the additional barrier loading induced by the normal stress on top of the dead zone according to basic principle in geotechnics. The computation of the additional loading on the barrier resulting from the shear stress (drag) raises more questions. The first question concerns the deposit length to consider for computing the equivalent shear-induced force applied on the deposit surface.  Kwan (2012) considers the length of the active wedge, which depends on the deposited material friction angle.  However, there is no evidence that this angle is appropriate in such a situation (saturated material characteristics and loading configuration). Also, considering an active wedge implies that the deposited material as a fiction angle, while considering a K value of 1 basically implies that the material is non frictional. The second question concerns the distribution of the induced load on the barrier. Kwan (2012) assume that the barrier is exposed to additional loading over its whole height. Others suggest it concerns only the top of the barrier.  This latter assumption seems to be confirmed by the fact that overflow induced significant increase in cable elongation at the top of the barrier. Because there are too many assumptions and open questions, the authors decided not to compute the additional barrier loading due to drag. They only computed the additional loading due to the normal stress induced by the flow on the deposit, in addition to the loading due to the deposit.

[Figure]

Back to the suggestion made, it appeared that the drag force computed based on equation 14 in Ng et al (2022) and considering the tests conditions and results presented in this manuscript, leads to drag force value exceeding 100 N by far. By comparison, the force acting on the barrier was in a 80-120 N typical range. Note is made that the measurement concerned a static load (at rest), while the drag is a transient load which vanishes to zero.

In the end, the Discussion section was enriched accounting for this comment and the provided reply.

**Technical comments**

1.     Pg 4, Para 90: Replace "recipe" with "flow composition"

Authors' response: Thanks for the suggestion. Its was accounted for.

2.     Pg 6, Para 120: Correct "lenght"

Authors' response: Thanks for pointing out this typo.

3.     Pg 8, Para 170: Please check and correct repetitive words

Authors' response: Thanks for this comment which was accounted for (in fact it concerned line 174)

4.     A thorough proofreading of the manuscript is recommended, as many grammatical errors and inconsistent terminologies are noted. Grammar needs to be checked and corrected- 280, 405 etc.

Authors' response: Thanks for the suggestion. The authors agree and made many improvements to the text

5.    Quality of the plates used in Fig 8 is low. Ensure that the plates are consistent with each other to enable proper comparison. Annotate the plates to indicate scale.

Authors' response: Thanks for the suggestion. It was accounted for.

6.    Fig 9: Adjust the axes to include upper and lower bound values.

Authors' response: Thanks for the suggestion. It was accounted for.

---

## Author Response (AR1)

Prof. Roberto Greco and the *Natural Hazards* and *Earth System Sciences* editorial board

Prof. Dr. Huo Miao (Sichuan Agricultural University)

Dr. Stéphane Lambert (Univ. Grenoble Alpes)
Dr. Guillaume Piton (Univ. Grenoble Alpes)

Objet: Submission of the revised version of research article EGUSPHERE-2024-3575

08 August 2025

Dear Prof. Roberto Greco,

Please find enclosed a revised version of the paper entitled "Capture of near-critical debris flows by flexible barriers: an experimental investigation" for submission in *Natural Hazards and Earth System Sciences*.

We carefully revised the paper according to the three referee comments and to your final appraisal. We submitted online on the *EGUsphere* website detailed responses to each reviewer. "Prof. Greco, your last assessment of their questions and our responses stressed that some of our responses to their comments were possibly not clear enough and that we should be careful to revise the paper making sure that these questions are addressed. We concur with this assessment and we revised again the paper to make it sure. Your full assessment is pasted at the end of this letter, but you more particularly stressed that "From your replies, I have noticed that in many cases it seems that you prefer sticking to your original text when the referee found it not completely clear (e.g., comments 1 and 5 from referee #1, and comments 1, 2, 4 and 5 from referee#2)."

In essence, these comments concern several points to which we would like to address new responses as we brought new modifications in the paper body:

• Comments 1 of referee #1 and 1 and 4 of referee #2 raise the question of the novelty and relevance of our study that focuses on near-critical debris flows (i.e. Froude ~1). The referees cite several papers that address much wider ranges, more precisely higher values, typically 1-7 or 1-10. Then, we are asked to justify why we think innovative, important and useful to restrain the focus on narrower ranges.

We would like to remind, that Prof. Johannes Hübl of BOKU Univ. in Vienna (AUT), a worldwide respected professor in the field of debris flows, in his paper Hübl *et al.* (2009)¹ analysed researches on impact forces and found that most experiments and simulations were performed on these high values of Froude number. Having field data monitoring not overlapping with most of these studies, he stressed that "Concluding one can state, that models are developed of an input data range which does not comply with field data. This is a systemic error." Since then, various papers presenting precise field measurements of Froude number of torrents in France, Switzerland and Italy allow us to fully concur with this statement. These articles are cited in the Introduction section.

The recent papers suggested by the referees are of value but have the same approach that was pointed by Prof. Hübl: exploring very high values of Froude number. This point is explained at length in almost all the Introduction. We are a bit surprised and we do not understand how we could justify more than through all our Introduction that a large share of the existing literature

<sup>1 Hübl J, Suda J, Proske D. 2009. Debris flow impact estimation steep slopes. In Proceedings of the 11th International Symposium on Water Management and Hydraulic Engineering. 1–5 pp.

explores a very wide range of phenomenon. Our aim was not to explore in detail a range of process going from slowly creeping surges of Fr<<1 to rocket-speed surges with Fr>7-10 for example with a new sensor, a new flume or with a more sophisticated model, again the same range of theoretical processes. This is a valuable line of research that is maybe followed by the reviewers but we decided not to follow this one. On the contrary our aim was to conduct a detailed investigation focusing on debris flows with a narrower but realistic (in the Alpine context at least) range of Froude number, to define realistic process type and to test them in the lab. The novelty is that this zoom, plus the use of a realistically behaving flexible barrier (also a novelty, done for the first time in small-scale model to the best of our knowledge, this is also mentioned and more stressed in the new version), enables to capture and describe flow – obstacle interactions with details not previously known. We did our best to revise one more time the Introduction to make it even more clear. We hope it will answer the questions raised by the referees and that we will not be asked again to explore a range of phenomenon that doesn't concern debris flows observed in many Alpine environment torrents.

• Comment 2 of referee #2, as well as less directly comment 5 of referee #1, point the absence of analysis of the impact force and ask why we focus on the force at rest. The practical justification was not detailed in the previous paper version. We added explanations of this limitation in the second paragraph of Section 2.5 of the revised version (underlined in the quotation hereafter): "Indeed, obtaining a precise measurement of the peak force was not possible due to some technical limitations with the system (slight smoothing of the force signal inside the sensor). A few independent tests with another sensor performed after the whole series let us think that the uncertainty on the peak force in our measurements is in the range 10-20%, sometimes possibly more. As a consequence, we decided not to focus on these uncertain peak force measurements to rather analyse force at rest for which the smoothing does not induce an uncertainty."

We would had love to share with everybody and publish nice peak force measurements with our realistic flows and barriers. So many research papers focus on this question, and comparing our data with these would have been a real achievement. Because our sensors had a bias, rather than to publish biased values, we decided to focus the analysis on other interesting points that usually attract less attention by researchers. We hope that the Associate Editor and the referees will understand that the paper is still of values even though it does not address the main scientific question that most of this literature usually addresses, with more precise and sophisticated methods than we could deploy in the work.

- Comment 5 of referee #1 was also questioning how exactly the force on the barrier Fb was computed. The Appendix of the paper was revised to make it fully clear. Sorry that the equations were not provided extensively.
- Finally, Comment 5 of referee #2 points that "The flexible barrier is open-type with the net allowing the pass of debris flow materials. But this part has not been discussed. I think it is also important for loading response of barrier in addition to the overflow process." We do not really understand how a real scale flexible barrier could not be open-type. We suppose that the referee meant to ask whether a large share of the interstitial fluid passed through the mesh opening. We agree that this question was maybe not detailed enough and so new elements were added to Section 2.3 describing the barrier. We hope it is now clearer.

We hope these responses demonstrate that we took the comments of the Associate Editor and Referees seriously. The associated revision of the paper shows that we profoundly rework and rewrote parts of the Introduction, of the Material and Methods as well as of the Results and of the Conclusion sections. In addition, the Discussion section was almost completely rewritten. We hope that *Natural Hazards and Earth System Sciences* will find some interest in this work and we are proud to publish in this journal.

Sincerely,

Prof. Dr. Huo Miao

Dr Stéphane Lambert

Dr. Guillaume Piton

PS: in addition to these elements, our responses to the reviewers' comments are available on the paper online discussion at <a href="https://doi.org/10.5194/egusphere-2024-3575-AC1">https://doi.org/10.5194/egusphere-2024-3575-AC1</a>, <a href="https://doi.org/10.5194/egusphere-2024-3575-AC2">https://doi.org/10.5194/egusphere-2024-3575-AC2</a> and <a href="https://doi.org/10.5194/egusphere-2024-3575-AC3">https://doi.org/10.5194/egusphere-2024-3575-AC3</a>

Last Editor Decision:

"Interactive discussion - 01 Jul 2025

Editor decision: Reconsider after major revisions (further review by editor and referees) by Roberto Greco Public justification (visible to the public if the article is accepted and published): Dear Authors,

based on the comments by three referees and on my personal assessment, I think that the manuscript needs major revisions before being reconsidered for possible publication. From your replies, I have noticed that in many cases it seems that you prefer sticking to your original text when the referee found it not completely clear (e.g., comments 1 and 5 from referee #1, and comments 1, 2, 4 and 5 from referee #2). As something that is not totally clear to a referee may be unclear also to the readers of the article, I invite you to carefully consider the comments received, even when you think that your text already contained what the referees suggested pointing out more clearly or explicitly.

However, I am confident that there is much room for improvement, and I look forward to receiving the revised version.

Best regards,

Roberto Greco"